# Vacillation, indecision and hesitation in moment-by-moment decoding of monkey motor cortex

Matthew T Kaufman[1,2,3]*, Mark M Churchland[4,5,6,7], Stephen I Ryu[1,8], Krishna V Shenoy[1,2,9,10]*

[1]Department of Electrical Engineering, Stanford University, Stanford, United States; [2]Neurosciences Program, Stanford University, Stanford, United States; [3]Cold Spring Harbor Laboratory, Cold Spring Harbor, United States; [4]Department of Neuroscience, Columbia University Medical Center, New York, United States; [5]Grossman Center for the Statistics of Mind, Columbia University Medical Center, New York, United States; [6]David Mahoney Center for Brain and Behavior Research, Columbia University Medical Center, New York, United States; [7]Kavli Institute for Brain Science, Columbia University Medical Center, New York, United States; [8]Department of Neurosurgery, Palo Alto Medical Foundation, Palo Alto, United States; [9]Department of Bioengineering, Stanford University, Stanford, United States; [10]Department of Neurobiology, Stanford University, Stanford, United States

*For correspondence:
mkaufman@cshl.edu (MTK);
shenoy@stanford.edu (KVS)

**Competing interests:** The authors declare that no competing interests exist.

**Abstract** When choosing actions, we can act decisively, vacillate, or suffer momentary indecision. Studying how individual decisions unfold requires moment-by-moment readouts of brain state. Here we provide such a view from dorsal premotor and primary motor cortex. Two monkeys performed a novel decision task while we recorded from many neurons simultaneously. We found that a decoder trained using 'forced choices' (one target viable) was highly reliable when applied to 'free choices'. However, during free choices internal events formed three categories. Typically, neural activity was consistent with rapid, unwavering choices. Sometimes, though, we observed presumed 'changes of mind': the neural state initially reflected one choice before changing to reflect the final choice. Finally, we observed momentary 'indecision': delay forming any clear motor plan. Further, moments of neural indecision accompanied moments of behavioral indecision. Together, these results reveal the rich and diverse set of internal events long suspected to occur during free choice.

## Introduction

The study of decision making in the brain is often limited by the need to average over repeated trials. Yet if the decision process differs on different trials, we may miss the very events that would be most enlightening. To detect such events requires a moment-by-moment (*Briggman et al., 2005*; *Yu et al., 2009*) readout of brain states on single trials (*Churchland et al., 2007*; *Afshar et al., 2011*).

To achieve such a moment-by-moment view in the monkey, we applied state-space analysis methods (*Shenoy et al., 2013*) to simultaneous recordings from dorsal premotor cortex (PMd) and primary motor cortex (M1)—brain areas centrally involved in preparing and producing the movements that effect decisions. When a monkey is instructed about an upcoming movement but not yet allowed to make it, neurons in these areas show changes in firing rate that reflect the properties of the upcoming reach, including direction (*Evarts, 1966*; *Tanji and Evarts, 1976*; *Weinrich and Wise, 1982*; *Godschalk et al., 1985*), distance (*Riehle and Requin, 1989*; *Messier and Kalaska, 2000*), speed (*Churchland et al., 2006a*), and curvature (*Hocherman and Wise, 1991*; *Pearce and Moran, 2012*).

**eLife digest** Some decisions are easy to make. We know almost immediately what outcome we want to achieve and what actions are required to do so. But other decisions involve more deliberation: there may be more factors to consider or more at stake, or the best course of action may simply not be immediately apparent. Under such circumstances, we may hesitate, waver or even change our minds completely.

To date, the majority of experiments that have explored the neural basis of decision making have been unable to detect the test subjects changing their mind as they made their decision. Instead, those experiments have measured the average brain response over multiple decisions because they lacked the power to detect what happened on each individual decision. Kaufman et al. have now addressed this issue by training two monkeys to perform a decision-making task that encourages wavering and changes of mind, and examining each decision one by one.

The monkeys learned to use their finger to trace a path to one of two targets on a screen to earn a reward. As the monkeys performed this target-selection task, arrays of electrodes recorded the activity of two brain regions that are involved in the planning of movements. These signals reliably predicted which of the two targets the monkey was favouring, several hundred milliseconds before the monkey was instructed to start moving his finger. Moreover, these patterns of neural activity reflected whether the monkey responded immediately or hesitated, made a firm choice or wavered, or stuck to his original choice or changed his mind.

While behaviours such as hesitation and wavering feel intuitively familiar, this is the first time that they have been observed at the neural level. However, it was not clear whether the two regions of the brain studied in the experiments were responsible for making the decisions about target selection, or if the activity in these areas reflected a decision that had been taken elsewhere in the brain. Nevertheless, the results indicate that the approach developed by Kaufman et al. allows researchers to follow aspects of the decision-making process as it happens, including those times when the monkey changes its mind.

PMd and M1 may (*Pastor-Bernier et al., 2012*; *Thura and Cisek, 2014*) or may not be involved in actually making decisions, but these areas are known to reflect the momentary decision state (*Thura et al., 2012*; *Thura and Cisek, 2014*) and therefore can be exploited as a window into the ongoing decision process. While studies of decision making typically focus on deliberative decisions based on noisy sensory evidence (e.g., *Britten et al., 1996*; *Shadlen and Newsome, 1996*; *Uchida and Mainen, 2003*; *Afraz et al., 2006*) or value optimization (e.g., *Sugrue et al., 2004*, *2005*; *Thura and Cisek, 2014*), we used a novel task in which the monkey was encouraged to decide quickly, yet often had time to change his mind if desired. Though recent studies have examined changes of mind driven by fluctuating sensory evidence (*Resulaj et al., 2009*; *Bollimunta et al., 2012*; *Insabato et al., 2014*; *Kiani et al., 2014*), the internal events that accompany truly free choices have largely remained hypothetical. Using single-trial methods, we were able to examine how free choices proceed despite the potential for this process to vary from trial to trial. In particular, we were able to test whether free choices generally proceed similarly to 'forced choices' in the motor system, determine how frequently the monkeys revised their initial motor plan, and provide insight into why some trials suffered slow reaction times (RTs).

## Results

### Task and behavior

We employed a novel decision-making variant of the previously described maze task (*Churchland et al., 2010*; *Kaufman et al., 2013*, *2014*). Monkeys (J and N) touched and fixated a central spot, then were shown two targets and four virtual barriers (*Figure 1A*). The monkeys were required to remain still and maintain eye fixation until a Go cue was given following a 0–1000 ms exponentially distributed delay (*Figure 1B*). Two barrier configurations were used: 'T-maze' (*Figure 1C*) and 'S-maze' (*Figure 1D*). In each configuration, two 'key barriers' appeared in one of three lengths making the nearby target easy (shown as dark gray in *Figure 1C,D*), hard (medium gray in *Figure 1C,D*), or

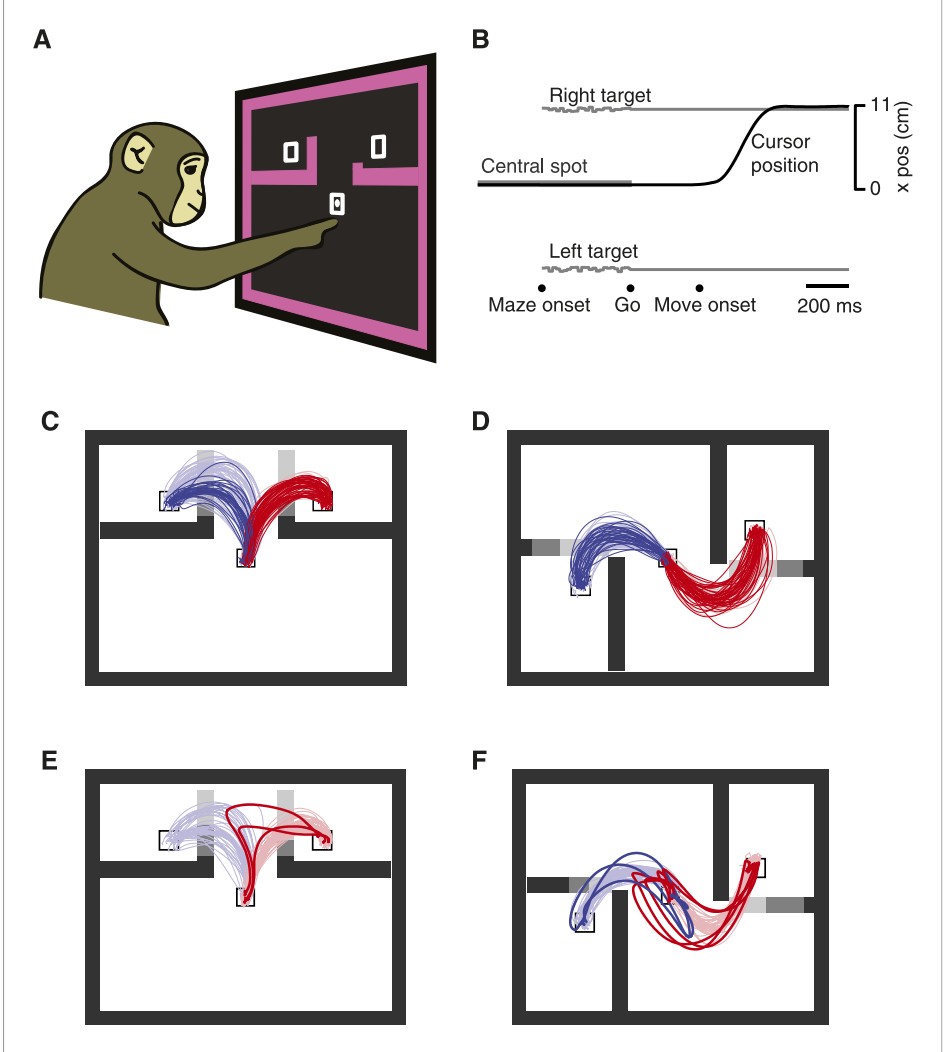

**Figure 1**. Decision-maze task. (**A**) Illustration of task setup. Two targets were presented along with four virtual barriers and a frame. The monkey performed the task with a cursor projected above his fingertip. Targets were rewarded equally. The cursor left a white trail on the screen. (**B**) Task timeline. (**C** and **D**) The two families of mazes used: 'T-maze' (**C**) and 'S-maze' (**D**). Key barriers could take one of three positions, making each target easy, difficult, or blocked (shown here as shades of gray). Reaches for trials with ≥300 ms delay shown. Faded colors, reach trajectories on forced choice trials; saturated colors, reach trajectories on free choice trials. (**E** and **F**) Overt changes of mind on free-choice trials with no barrier changes. Dataset J1 (**A–E**); dataset N3 (**F**).

The following figure supplement is available for figure 1:

**Figure supplement 1**. Overt changes of mind for the other five datasets (labeled on panels).

inaccessible (light gray in *Figure 1C,D*). Either key barrier could change difficulty at a random point in the trial (~58% of trials). At most one change was made per trial.

This task presented a continuum of situations, ranging from complete experimenter control ('forced choices', with only one target accessible), to moderate experimenter influence (e.g., a barrier that changed mid-trial might provoke a change of mind), to free choices with both targets accessible. This task is somewhat analogous to a driver choosing a lane. Often one attempts to pick a lane that will be most expeditious. The choice may be determined by the environment: for example, one lane is blocked by a large truck and the other is nearly empty. On other occasions the choice is more ambiguous. Both lanes may have some cars or both may be clear. We are all familiar with the experience of rapidly weighing these choices, and with the vacillation and hesitation that can result.

**Table 1.** Choice probabilities

| | p(left) easy\|easy | p(left) hard\|hard | p(left) easy\|hard | p(left) hard\|easy | Δp for left-biasing change | Δp for right-biasing change |
|---|---|---|---|---|---|---|
| J1-T | 0.66 | 0.02 | 0.95 | 0.01 | 27% | −38% |
| J1-S | 0.16 | 0.57 | 0.66 | 0.17 | 32% | −19% |
| J2-T | 0.92 | 0.00 | 0.92 | 0.00 | 20% | −44% |
| J2-S | 0.23 | 0.99 | 1.00 | 0.06 | 45% | −38% |
| N1-S | 0.98 | 1.00 | 1.00 | 0.20 | 37% | −37% |
| N2-S | 0.79 | 0.83 | 0.88 | 0.70 | −4%* | −15% |
| N3-S | 0.79 | 0.54 | 0.79 | 0.45 | 18% | −16% |

The first four columns show the probability that the monkey chose the leftward target given the particular barrier configuration. 'Easy|hard' means that the barrier configuration was easy for the left and hard for the right. Trials included for these columns had no barrier changes. The Δp values show the change in p(left) when a trial presented a free choice throughout, but the difficulty of a key barrier changed during preparation (from 100 ms after maze onset to 50 ms after Go). These are separated by whether the change favored the left target (the left barrier became easier or the right harder) or the right target. Because dataset N2-S did not have consistent behavioral effects on trials with a change in barrier difficulty (starred entry), N2-S was not used for one analysis of biasing change trials.

Our task is roughly analogous: often the stimulus does not fully specify the right answer, leaving a choice to be made.

In agreement with this idea, the monkeys' choices statistically reflected the relative difficulties of the two options (*Table 1*), similar to previous observations (*Cos et al., 2011*). As expected, choices were also influenced by barrier changes during the delay period (*Table 1*; dataset N2 showed imperfect responses to barrier changes, and was thus excluded from one analysis below). For example, if the leftward target became more accessible as the result of a barrier change then the probability of choosing left increased. While biases were common (much as they would be for drivers choosing lanes) both monkeys made reaches to both targets when given a choice, and reached more often to a target when it was easy than when it was hard. Finally, the monkeys occasionally 'changed their mind' mid-reach: they began reaching toward one target but performed a sharp turn and reached to the other target (*Figure 1E,F*, *Figure 1—figure supplement 1*). In most of these overt change-of-mind trials, both options were available when the monkey deviated from his original reach. Depending on the trial, these changes of mind were driven either by a barrier change that only altered the relative difficulties of the two options, or by some purely internal process. Thus, 'freedom of choice' was indeed exercised.

Trials were sorted post hoc to identify those relevant for the various analyses below. We first describe results from the ends of the choice spectrum (fully forced choices and fully free choices), then consider cases where barrier changes encouraged changes of mind. Finally, we examine how the neural state related to RT.

## Multi-unit responses during forced and free choices

All analyses were, by necessity, performed on the data collected within a single day: the population of recorded neurons, as well as any monkey's internal events and behavior, likely change from day to day. To provide the necessary statistical power to observe internal events on a single-trial basis, neural data were recorded using two 96-electrode arrays (in PMd and M1) per monkey. These recordings yielded 96–196 single- and multi-units per dataset. Since neurons were recorded simultaneously, we could neither optimize target locations for each neuron separately nor select for strongly responsive neurons. Instead, we selected target arrangements that evoked both possible choices and drove particularly robust population responses, which for some datasets yielded sufficient statistical power for good decoding of choice on single trials. This selection yielded seven datasets for analysis: two target arrangements (T and S mazes) for monkey J across 2 days (J1 and J2), and one target arrangement (S maze) for monkey N for 3 days (N1, N2 and N3).

Neural responses during forced-choice trials showed delay-period activity (after maze onset but before the Go cue) that rapidly (in 100–200 ms) reflected the available choice (*Figure 2A,B*).

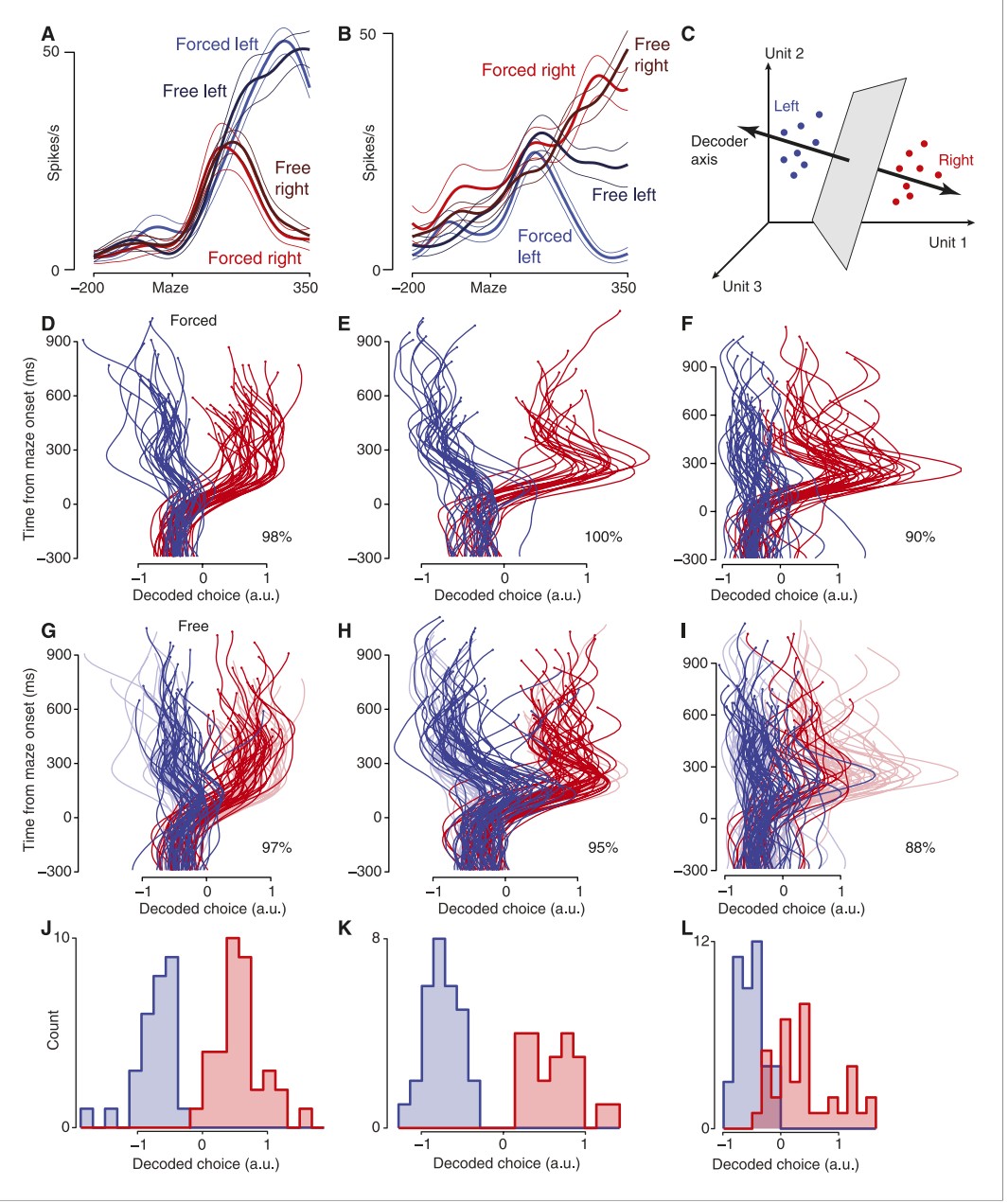

**Figure 2**. Firing rates and decoding. Blue represents eventual leftward reaches, red indicates eventual rightward reaches. (**A** and **B**) Responses of example units, dataset J2-S. Thick traces, mean; thin traces, s.e.m, Maze, maze onset. Time in ms. Selectivity for left vs right movements was common, and responses were almost always similar for forced and free reaches (**A**). Less commonly, forced and free evoked somewhat different responses (**B**). (**C**) Schematic of decoder. Each dot represents neural state in a window of time on a single trial. (**D**–**F**) Decoded choice plots for forced-choice trials, generated by leave-one-out cross-validation. Percentages show fraction correct classification. Datasets J2-T (**D**), J2-S (**E**), N1-S (**F**). Small dots are at last decoded time point. (**G**–**I**) Decoded choice plots for free-choice trials (saturated colors) with forced-choice trials shown for context (faded colors). Datasets same as **D**–**F**. (**J**–**L**) Cross-validated decoded choice at final point for forced-choice trials. Datasets same as **D**–**F**.

The following figure supplements are available for figure 2:

**Figure supplement 1**. PSTHs for more example units.

**Figure supplement 2**. Decoding forced and free choices.

In order for this to occur, the monkey needed to visually parse the arrangement of barriers and targets, determine which target was accessible, and form a motor plan that would reach the target and avoid the barriers. A key question is whether the neural activity during free choices reflects the eventual movement in a similar way and with a similar time course. This is not a given: during free choices, the preparatory activity might maintain its baseline state, achieve an intermediate state to make a change of mind 'easier' (*Cisek and Kalaska, 2005*; *Fleming et al., 2009*; *Ifft et al., 2012*), might develop more slowly, or might vacillate between the choices continually.

To answer this question, we segregated free choice trials by whether the monkey eventually reached left vs right (*Figure 2A*, *Figure 2—figure supplement 1*; possible change-of-mind trials were excluded, see 'Materials and methods'). For most neurons, the average response only depended on which reach was made; responses were similar regardless of whether a choice was forced or free. Interestingly, for most neurons, the time course of the neural response was nearly identical for forced and free choices. This implies that the monkey formed a motor plan quickly when presented with a free choice: that is, the time to make an initial movement selection for free choices did not take appreciably longer than the time to determine which target was accessible for forced choices. For a smaller fraction of neurons, free-choice trials exhibited selectivity that was weaker than for forced-choice trials (e.g., *Figure 2B*). However, neurons consistently 'preferred' the same target in free-choice trials as in forced-choice trials (102 of 104 significantly tuned units maintained their preference), and neurons' trial-averaged responses during the delay strongly correlated for forced and free choices (mean r = 0.76, p < 0.001 for 6/7 datasets, p < 0.01 for N3-S; sign test; 'Materials and methods'). In sum, activity on free choice trials was consistent with the interpretation that initial choices were typically made rapidly. Below, we consider whether some of those choices might spontaneously waver or reverse.

## Decoding the moment-by-moment population response

To analyze the population response on single trials, we trained a linear decoder using the forced-choice trials ('Materials and methods'). This decoder was simply a weighted sum of the neurons' smoothed spike counts (*Figure 2C*). Empirically, the decoder chose to exploit the activity of most neurons, with the decoder's weights distributed over units approximately as a double-exponential distribution (not shown). The same decoder was used across a set of delay-period time points: from 300 ms before maze onset to 200 ms before movement onset. Though it made decoding more challenging, the time invariance of the decoder was essential to ensure that any changes of mind we observed were solely due to changing neural activity.

This decoder performed well on forced-choice trials, as verified by leave-one-out cross-validation. Each trial's decoded choice over time is represented by a single trace, with the horizontal coordinate representing the decoded choice and time unfolding upward (*Figure 2D–F*; *Figure 2—figure supplement 2A*). Trials culminating in a leftward reach are shown in blue, while trials culminating in a rightward reach are shown in red. At the final time point (200 ms before movement onset), the decoder performed correctly on 99% of forced-choice trials for monkey J and 94% of trials for monkey N (p < $10^{-6}$ for each dataset assessed individually, 'Materials and methods'). Conveniently, errors were all 'small', lying just on the wrong side of the classifier boundary (three datasets shown in *Figure 2J–L*; others in *Figure 2—figure supplement 2C*).

## Free choices

This same decoder (trained on forced-choice trials) typically generalized well to free-choice trials. For the datasets containing at least 1000 trials, at the final time point (200 ms before movement onset) the choice was decoded correctly on 96% of trials for monkey J and 88% of trials for monkey N. Free choice decoder performance was less accurate for the two datasets with fewer trials (N2-S and N3-S), at 80%. Neural data from all the long-delay free choice trials (including four barrier configurations), in which neither target was blocked and in which no barrier changes occurred, are shown in *Figure 2G–I* (saturated traces, forced-choice trials shown as faded traces for context; see also *Figure 2—figure supplement 2B*; p < $10^{-6}$ for all datasets). As expected, decoding was unsuccessful before maze onset, for either forced or free choices. The slightly lower decoder performance on free choice trials was not due to any systematic shift in neurons' preferences: retraining the decoder using free-choice trials did not improve performance on free-choice trials (96% and 77% for monkeys J and N, assessed via leave-one-out cross-validation). Thus, in PMd and M1 the neural events during free choices closely resemble those during forced choices that result in the same movement. Rather, the lower performance

of the decoder may reflect some additional complexity during free choices, such as last-moment changes of mind. The decoder is not given access to the data from shortly before movement onset, so any such last-moment changes of mind will necessarily be missed.

It is immediately clear from these free-choice plots that there are some trials in which the decoder first indicated a plan toward one target, then switched to the other target (traces crossing the midline/decoded value of 0 in *Figure 2G–I*). If the decoder is reliable at many time points, then these crossings would be evidence for covert changes of mind (*Horwitz and Newsome, 2001*; *Resulaj et al., 2009*; *Bollimunta et al., 2012*; *Kiani et al., 2014*; *Thura and Cisek, 2014*). Below, we present evidence that these early decoder values are indeed reliable, then quantify these changes of mind.

## Early decision activity relates to behavior

In order to search for fully covert processes such as vacillation (unprompted changes of mind), it is important to determine the reliability of the decoder across time points. We did so in three ways: by exploiting our distribution of delays, by examining the decoded choice on trials where changes-of mind must occur a large fraction of the time, and by examining the RTs for trials with a particular set of events.

For the first method, we considered trials with relatively short delays. Short-delay trials allow us to verify the accuracy of the decoder at various time points, since the monkey could not know in advance how long the delay would be for any given trial. If these early time points are decoded accurately, then we can rely on the short-latency decoded points for long-delay trials as well. Using these trials, we found that decoder accuracy plateaued ~200–250 ms after maze onset (*Figure 3A*).

Second, we wished to confirm that our readout could reflect the monkeys' choices at fast timescales. To do so, we examined 'free-to-forced' trials, in which one of the available choices became inaccessible due to a barrier change. Since we randomly chose which side to make inaccessible, on roughly half of such trials the monkey should have already fortuitously chosen the 'lucky' (unchanging) side. For the other 'unlucky' half of trials, we expect that the monkey must re-plan. This can be clearly seen in the decoded choice: when the barrier change occurred mid-delay (at least 100 ms after maze onset but at least 50 ms before the Go cue), as expected there were large swings in the decoded choice on about half the trials following the change event (*Figure 3B,C*, *Figure 3—figure supplement 1*; dots indicate barrier change). These large swings in the decoded choice contributed to strong decoder performance on free-to-forced trials: using the final decoded time, 91% of these trials were correctly decoded for monkey J, and 94% for monkey N. Thus, it was possible for the decoder to detect transient choices.

Third, we hypothesized that our readout of the internal choice reflected behaviorally important aspects of the neural state. The free-to-forced trials considered above presented barrier changes during the delay, which gave the monkey time to re-plan. In contrast, when barrier changes occurred around the time of the Go cue, we would expect that the monkey must spend time re-planning and thus have a slower RT. This was the case in our data. We considered free-to-forced trials in which the barrier change occurred around the time of the Go cue (no more than 50 ms before), and segregated them based on the decoded choice at the time of the barrier change. If the decoded choice indicated a plan to the unchanging side, the RT was faster than if it indicated a plan to the now-blocked side (*Figure 3D,E*, median difference 68 ms for monkey J, p = 0.009; median difference 24 ms for monkey N, n.s., Mann–Whitney *U* test).

## Spontaneous changes of mind (vacillations) are present in the neural activity

Do monkeys ever revise their choices? We hypothesized that monkeys would quickly make an initial choice, then occasionally revise it when given more time to consider. For fully free choices, there is no reason this must happen, yet intuition suggests that a person would occasionally vacillate when presented with two viable options (e.g., a driver choosing a lane). Indeed, in both monkeys a modest number of trials exhibited such spontaneous vacillations. This was apparent above in *Figure 2G–I* (and *Figure 2—figure supplement 2B*). Using a conservative algorithm ('Materials and methods'), we identified trials in which the decoder initially produced a value indicating one choice, then underwent a large swing to indicate the opposite choice. These trials are shown in *Figure 4A,B* (saturated traces; see also *Figure 4—figure supplement 1* for vacillation and non-vacillation trials plotted separately, and *Video 1* for two examples). Using these same conservative criteria, 13%

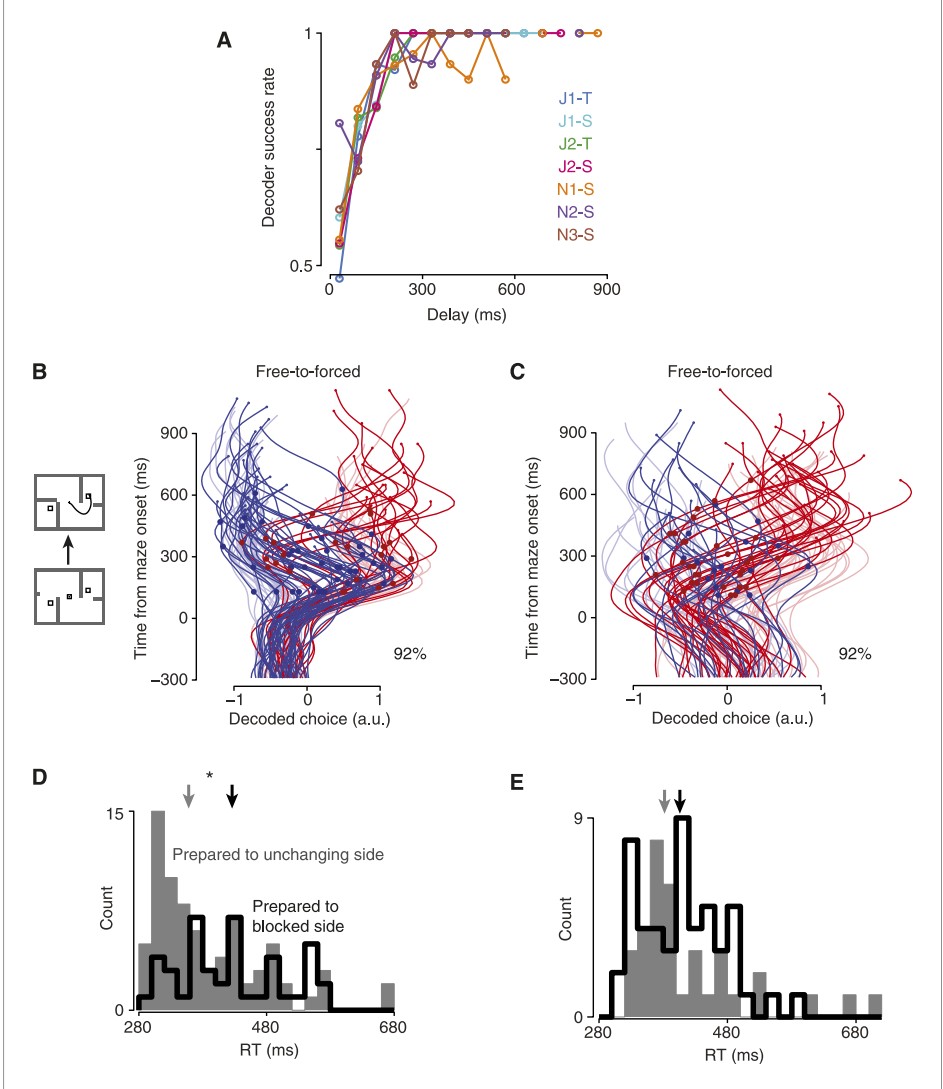

**Figure 3**. Validating the decoder. (**A**) Decoder performance vs time, forced-choice trials. (**B**) Maze icons illustrate free-to-forced trials. Saturated colors show the decoded choice for free-to-forced trials, which began as free but became forced when a barrier changed during the delay epoch. Faded colors show forced-choice trials for context. Large red dots indicate time of changes that made the rightward target the only available option; blue dots, leftward. Percentage indicates the fraction of trials for which the final decoded time point matched the monkey's choice. Dataset J2-S. (**C**) Same for dataset N3-S. (**D**) RT distributions for free-to-forced trials in which the barrier changed around the time of the Go cue. Black, trials where the monkey initially prepared a reach to the now-blocked side; gray, to the unchanging side. Data for monkey J pooled. Arrows, medians. (**E**) Same for monkey N.
The following figure supplement is available for figure 3:

**Figure supplement 1**. Decoded choices for free-to-forced trials.

(3–24% range over datasets) of free-choice trials exhibited such spontaneous vacillations, compared to only 2% (2–4%) of forced-choice trials (forced-choice assessed using leave-one-out cross-validated decoding). Vacillation trials were more common among free-choice trials than forced-choice trials in every one of the 7 datasets ($p < 10^{-9}$ pooled; $\chi^2$ 2 × 2 contingency test; *Figure 4I*, left). This difference in frequency also reached statistical significance for 5/7 datasets when considered individually ($p < 0.05$, $\chi^2$ 2 × 2 contingency test). To our knowledge, observations of such spontaneous vacillations when presented with a free choice—without changing external evidence—have not previously been described.

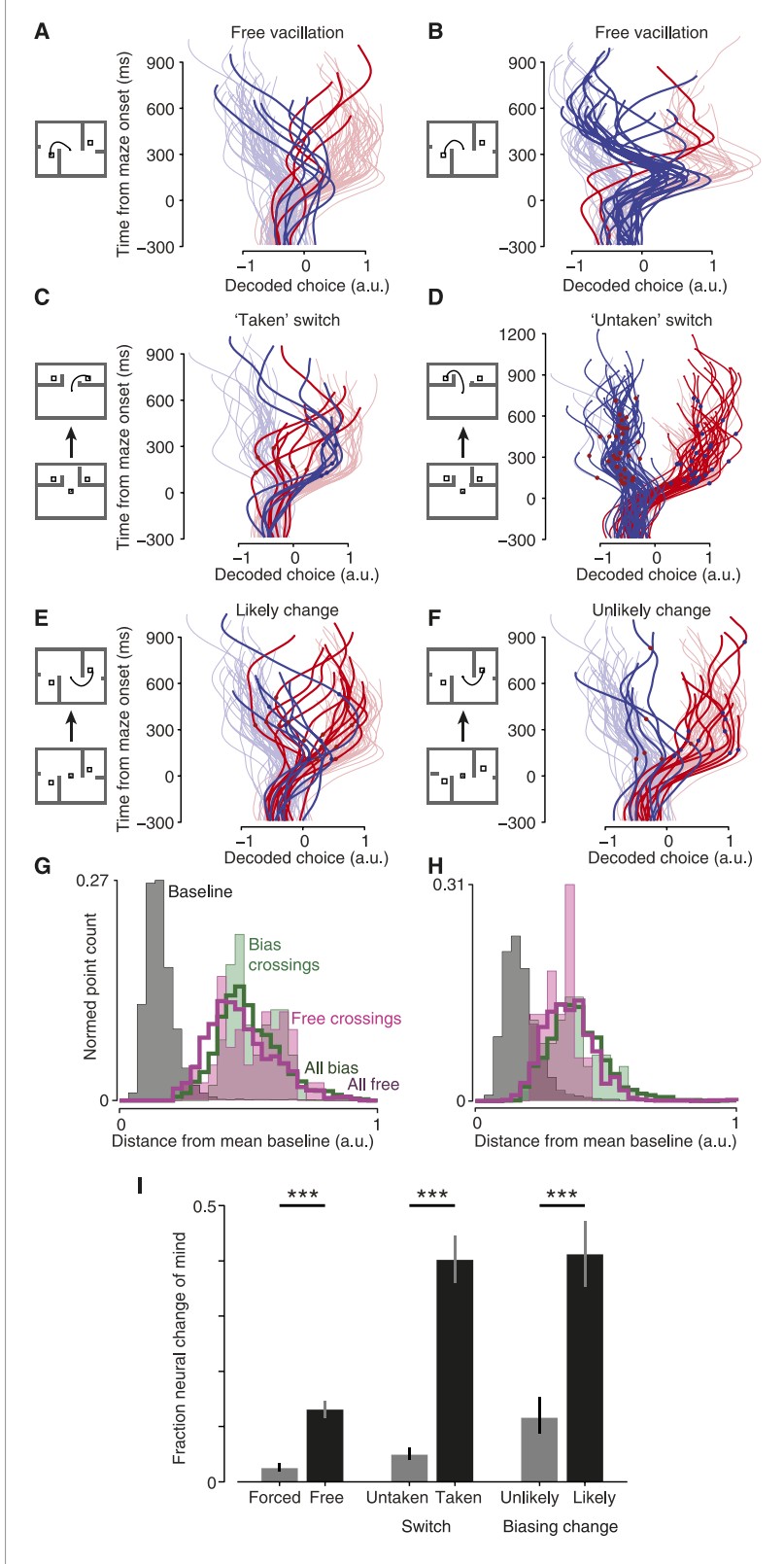

**Figure 4**. Covert changes of mind. Maze icons indicate situation for adjacent decoded choice plots (saturated colors), with faded colors showing forced-choice trials for context. (**A** and **B**) Apparent 'vacillation' on free choice trials. Datasets J1-S (**A**) and J2-S (**B**). (**C** and **D**) Encouraged switch trials, which began as forced but became free when a barrier changed mid-delay. Red dots on traces indicate time of changes that made the rightward target

*Figure 4. continued on next page*

*Figure 4. Continued*

more attractive; blue dots, leftward more attractive. J2-T. (**C**) Trials where the monkey chose the newly-available target. (**D**) Trials where the monkey chose the always-available target. (**E** and **F**) Biasing change trials. These trials were free throughout, but the difficulty of one side changed mid-delay. (**E**) Trials where the monkey reached to the target that was initially more difficult, and thus likely changed his mind. (**F**) Trials where the monkey reached to the target that was initially easier, and thus likely retained his initial decision. Dataset J1-S. (**G** and **H**) Distance from mean 'baseline state' (−300 to −40 ms from maze onset) for different trials and time epochs: during baseline (gray), during free-choice vacillations around the time of the change in the decoded choice (filled pink), during biasing-change trials around the time of the change in the decoded choice (filled green), during all times for free (hollow pink) or biasing-change (hollow green) trials. J2-S (**G**) and N1-S (**H**). (**I**) Probability that a trial of the given type exhibited a neural change of mind (a large change in the decoded choice during the delay period). See 'Results' for details. Forced vs free and untaken vs taken switch, $p < 10^{-9}$; unlikely vs likely change, $p < 10^{-4}$ ($\chi^2$ 2 × 2 contingency test). Error bars show Wilson binomial confidence intervals equivalent to 1 s.e.m.

The following figure supplements are available for figure 4:

**Figure supplement 1**. Free choice vacillation.

**Figure supplement 2**. Example vacillation trials with choice decoded using PMd and M1 separately.

**Figure supplement 3**. Decoded choices for taken switch trials (left column) and untaken switch trials (right column).

**Figure supplement 4**. Biasing changes in the barriers encouraged changes of mind.

**Figure supplement 5**. Neural states during changes of mind typically did not resemble the baseline state.

We performed two additional controls to aid in interpreting these apparent vacillations. First, we wished to verify that the activity in PMd and M1 agreed. For the 5 datasets that could be decoded most accurately (all four J datasets and N1-S), we considered the data from PMd and M1 separately. Specifically, we asked whether the decoded choice from these two areas agreed at time points at least 160 ms into the delay period; this time point was chosen to be late enough that the decoded choice reflected a real choice and not just noise ('Materials and methods'). Despite the much higher noise levels involved when splitting our data into two (not necessarily equal) parts, the results from decoding PMd agreed with the results from decoding M1 with a similar reliability as expected from how well each decoder agreed with the behavior. This was true when considering all trials (*Table 2*) or free choice trials alone (*Table 3*). Moreover, we could examine vacillation trials specifically. On the same trials, we could often see similar vacillations when decoding PMd and M1 separately (*Figure 4—figure supplement 2A–C*), though this was not always the case (*Figure 4—figure supplement 2D*).

Second, we addressed the concern that some of these vacillations might be related to the tendency for the baseline state (before maze onset) to produce a 'leftward' decoded choice in most datasets. To check for this possibility, we performed a simple control. For each vacillation trial, we considered the last time the decoded choice switched sign. We counted how many times it was left-to-right, and how many times it was right-to-left. In 3/7 datasets, left-to-right switches were more common; in 3/7, right-to-left were more common; in 1 dataset they were tied. It is not entirely clear why at baseline the decode choice tended to register as 'leftward', but it does not appear that this baseline bias affected our ability to identify vacillations.

## Changes of mind in a changing environment

On some trials, we changed the difficulty of one option mid-trial to invite (but not force) the monkey to change his mind. We hypothesized that the monkey would sometimes change his mind on such trials, to be responsive to a changing environment.

The most straightforward of these trials was the 'encouraged switch' (forced-to-free) case. These trials began as forced, then a barrier changed difficulty during the delay to provide a free choice. On some trials the monkey reached to the always-accessible option ('untaken' switch trials). On other trials ('taken' switch trials) the monkey reached to the side that had changed, presumably reflecting a change of mind

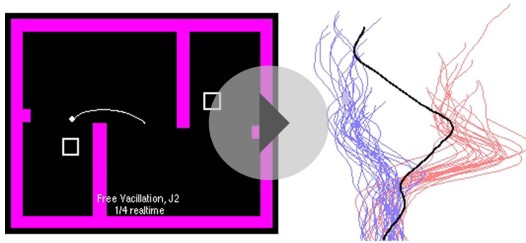

**Video 1.** Two example free choice vacillation trials. The left panel shows the display that the monkey saw, plus descriptive text. The right panel shows the decoded choice trajectory for the same trial (black), and the decoded choice trajectories for forced choice trials for context (blue for left, red for right). Video is presented at 1/4 real-time speed.

from the initial option to the newly-accessible target. Indeed, for taken-switch trials the decoded choice over time often reflected the change of mind well before movement onset (saturated traces in *Figure 4C*; *Figure 4—figure supplement 3*; *Video 2*). For these taken-switch trials, 40% exhibited a large change in the decoded choice (15–75%; assessed as for vacillations; 'Materials and methods') in the period from 160 ms after maze onset until 200 ms before movement onset. The remaining taken-switch trials presumably involved switches in the internal state that occurred after the analyzed period (i.e., just before or as the monkey began reaching), consistent with recent work showing that planning and re-planning can be rapid (*Ames et al., 2014*; *Wong et al., 2014*). On untaken-switch trials, the decoded choice was generally steady, as expected (*Figure 4D*). Only 5% (0–17%) of untaken-switch trials exhibited such a large change in the decoded choice. In all 7 datasets, large changes in the decoded choice were more common for taken-switch trials than untaken-switch trials (p < 10⁻⁹ pooled; p < 0.02 for 6/7 datasets assessed individually, N3-S n.s.; $\chi^2$ 2 × 2 contingency test; *Figure 4I*, center). Thus, changes in the decoded choice were common for conditions where they were expected to be present, and uncommon for conditions where they were expected to be rare. This result provides further validation of the decoder.

We then examined a case where decisions were more nearly free yet still under experimenter influence: 'biasing change' trials. These trials began with a free choice, with one side hard and the other easy. During the delay, the hard key barrier changed to become easy as well. Behavioral statistics indicate that these barrier changes successfully induced changes of mind on many trials (*Table 1*). In particular, there were many 'likely change' trials: for example, a trial where the rightward target had been hard, became easy, and was eventually chosen. Yet behavior alone cannot tell us *which* of those trials involved internal changes of mind: the monkey may have chosen the hard target from the start, as sometimes occurred during free choices. At the neural level, large changes in the decoded choice were frequently apparent: they comprised 41% (29–50%) of such likely change trials (saturated traces in *Figure 4E*; *Figure 4—figure supplement 4*; *Video 3*). Conversely, on trials where the monkey eventually reached to a target that was easy throughout, changes of mind were

**Table 2**. Decoding using PMd and M1 separately, all trials

| | Forced-choice decoder performance (leave-one-out) | | | Decoder agreement | | |
|---|---|---|---|---|---|---|
| | Combined | PMd | M1 | PMd-combined | M1-combined | PMd-M1 |
| J1-T | 0.99 | 0.99 | 0.90 | 0.92 | 0.85 | 0.80 |
| J1-S | 1.00 | 1.00 | 0.80 | 0.93 | 0.86 | 0.81 |
| J2-T | 0.98 | 0.97 | 0.80 | 0.96 | 0.87 | 0.86 |
| J2-S | 1.00 | 1.00 | 0.85 | 0.98 | 0.84 | 0.83 |
| N1-S | 0.90 | 0.84 | 0.95 | 0.93 | 0.72 | 0.67 |

Statistics for decoding choice from PMd and M1 data separately. Decoder performance refers to the fraction of trials for which the final decoded point agreed with the target that the animal reached to. This was assessed on forced-choice trials with delays of at least 300 ms, using leave-one-out cross-validation. Decoder agreement refers to the mean fraction of time points per trial for which the decoded choice was the same using the two datasets indicated. For this statistic, all successful trials with delay periods of at least 300 ms were included. 'Combined' refers to the decoder trained using both PMd and M1 data.

**Table 3**. Decoding using PMd and M1 separately, free choice trials

| | Free-choice decoder performance | | | Decoder agreement | | |
|---|---|---|---|---|---|---|
| | Combined | PMd | M1 | PMd-combined | M1-combined | PMd-M1 |
| J1-T | 0.94 | 0.86 | 0.77 | 0.90 | 0.83 | 0.77 |
| J1-S | 0.96 | 0.94 | 0.89 | 0.86 | 0.82 | 0.73 |
| J2-T | 0.97 | 0.96 | 0.82 | 0.96 | 0.82 | 0.80 |
| J2-S | 0.95 | 0.95 | 0.86 | 0.98 | 0.78 | 0.78 |
| N1-S | 0.88 | 0.82 | 0.79 | 0.91 | 0.72 | 0.67 |

Same as **Table 2**, but only considering free-choice trials with delay periods of at least 300 ms. The decoders were trained using forced-choice trials.

presumably uncommon: most likely the monkey immediately chose the easy target and maintained this plan. Accordingly, for these 'unlikely change' trials changes in the decoded choice were observed less frequently: only 12% (4–23%) of trials (**Figure 4F**). Likely-change trials exhibited large changes in the decoded choice more frequently than unlikely-change trials in all 6 datasets (N2-S excluded; see above discussion of **Table 1**), and this difference was highly statistically significant ($p < 10^{-4}$ pooled; $p < 0.05$ for 3/6 individual datasets, $\chi^2$ 2 × 2 contingency test; **Figure 4I**, right).

## Neural state during changes of mind

We wished to ask in greater detail what occurred during changes of mind. First, we asked whether these changes of mind required a return to the baseline state. To do so, we examined the distance between the neural state during changes of mind and the neural state before the maze onset (baseline). The changes of mind did not approach the baseline state for either induced changes (on biasing-change trials) or spontaneous vacillations (**Figure 4G,H**, **Figure 4—figure supplement 5**). That is, although we used a one-dimensional neural readout as a decoder, the neural state was multidimensional and in those other dimensions the neural state did not pass through baseline during changes of mind. This is consistent with previous findings about the neural response when a single target jumps from one location to another (**Archambault et al., 2009**, **2011**; **Ames et al., 2014**).

Second, we asked whether there was a population signature during changes of mind. This is challenging because of the heterogeneity of event timing, but we could nonetheless ask whether individual units' firing rates were unusually high or low during these changes of mind. To do so, we compared firing rates during a time window around the change of mind with matched windows on similar non-change-of-mind trials ('Materials and methods'). For trials in which a change of mind was induced by a changing barrier, there was no systematic difference in firing rates relative to non-change-of-mind trials; units had higher-than-average and lower-than average firing rates with similar frequencies (30% were higher, 26% lower, 44% in the normal range; $p = 0.16$ pooled over datasets). However, during vacillation trials, firing rates tended to be slightly low relative to non-vacillation trials: 37% of units had lower-than-average firing rates during the vacillation, while only 27% were higher than average ($p = 0.04$ for J; $p = 0.009$ for N; Z-test for proportions; remaining units were in the normal range). This suggests that changes of mind induced by outside events may represent a slightly different process from spontaneous vacillation. However, vacillations occurred earlier on average than shifts in choice due to barrier changes, so this firing rate difference may reflect the time course of motor preparation rather than a difference between spontaneous vacillation and induced changes of mind.

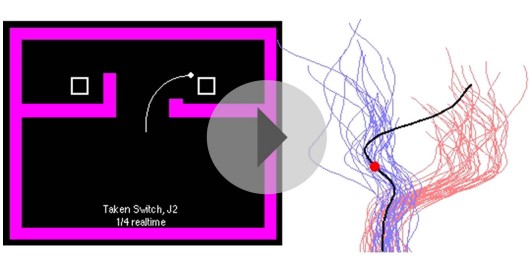

**Video 2.** Two example 'taken-switch' trials. These trials began as forced then became free, and the monkey reached to the newly-accessible target. A red or blue dot appears on the decoded choice trace at the time of the barrier change (dot color indicates which side the change favored). Video is presented at 1/4 real-time speed.

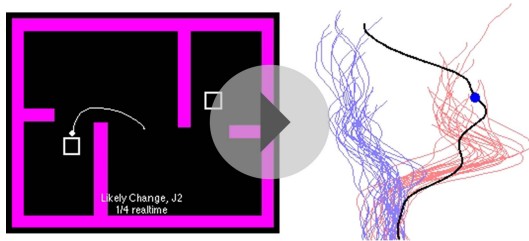

**Video 3.** Two example 'likely change' trials. These trials began with one side hard and the other easy, then a change made the initially easy side hard or the initially hard side easy. The monkey reached to the target that was not initially easy. Video is presented at 1/4 real-time speed.

## Hesitation and indecision

Finally, we exploited the single-trial view to examine two potentially informative but rare and variable circumstances—cases in which decision and/or execution were slowed. Focusing first on execution, we returned to untaken-switch trials. These correspond to a situation we have all experienced: at the last minute an option becomes available, but we choose not to exercise it. This last-moment offer nonetheless produced behavioral consequences: the monkey occasionally exhibited slowed RTs even though his choice was unaffected (*Figure 5A*, *Figure 5—figure supplement 1*). We hypothesized that this slowing might be due to a transient re-planning event. To our surprise, the decoded choice on these trials never wavered: the same decoded choice was always observed throughout each trial in all datasets (*Figure 5B*, *Figure 5—figure supplement 1*). This suggests that the slow RTs on these trials were not due to re-planning/vacillation, but instead involved a 'hesitation' to execute. That is, when presented with a new option, the monkeys hesitated to execute their seemingly still-valid motor plan.

Second, we examined an effect that corresponds to another intuitively common situation: the moment of indecision that can occur when faced with two good options. Such an effect is not inevitable (there is little advantage to indecision when both choices are good) and was not always observed. However, for one dataset the monkey did appear to undergo such moments of indecision: when a choice had to be made immediately upon maze presentation, the RT was often longer for two possible choices than for one choice (*Figure 5C*, dataset J2-S; median difference 56 ms, p < 0.001, Mann–Whitney *U* test). These indecisive moments were present for trials resulting in either left or right choices (RT distributions were generally similar for left and right). Although this interesting behavior was only present in one dataset, the ability to analyze individual trials allows us to identify neural correlates. We hypothesized that these slow RTs might be due to a slowly-developing initial motor plan or quick vacillation. In accordance with this idea, there was a strong relationship between a trial initially exhibiting an 'incorrect' decoded choice (relative to the eventual reach) and that trial having a slow RT (*Figure 5D*). Thus, this moment of indecision was distinct from hesitation, which did not have an obvious correlate in the decoded choice. As a control, we compared the datasets in which free-choice RTs were not slowed substantially (*Figure 5—figure supplement 2*); the neural effect was absent in these datasets, as expected. Thus, neural 'indecision' was observed only when there was behavioral indecision.

## Discussion

While changes of mind have been hypothesized to occur during decision making, and have even been observed when sensory evidence changes (*Horwitz and Newsome, 2001*; *Bollimunta et al., 2012*; *Kiani et al., 2014*; *Thura and Cisek, 2014*), little has been known about how decisions proceed during truly free choices. Here, we trained monkeys to perform a novel task that often presented two viable options with either answer resulting in the same reward. We then examined the dynamics of decision making using single-trial, moment-by-moment decoding of preparatory activity from PMd and M1. We found that motor plans for forced-choice and free-choice trials were typically similar and formed with similar speed. However, as suspected, these plans were sometimes revised later. These changes of mind were essentially never observed when the animal was presented with forced choices; this internal process was specific to free choices.

Several pieces of evidence support the validity of the changes of mind we observed. First, the decoder performed very well on forced choices, with nearly all decoder errors lying just on the wrong side of the threshold and not strongly favoring the opposite choice. These 'small' errors could not readily give rise to strong apparent changes of mind. Second, trials with short delay lengths effectively acted as probes of the decoded choice at different time points. These probes permitted us to verify

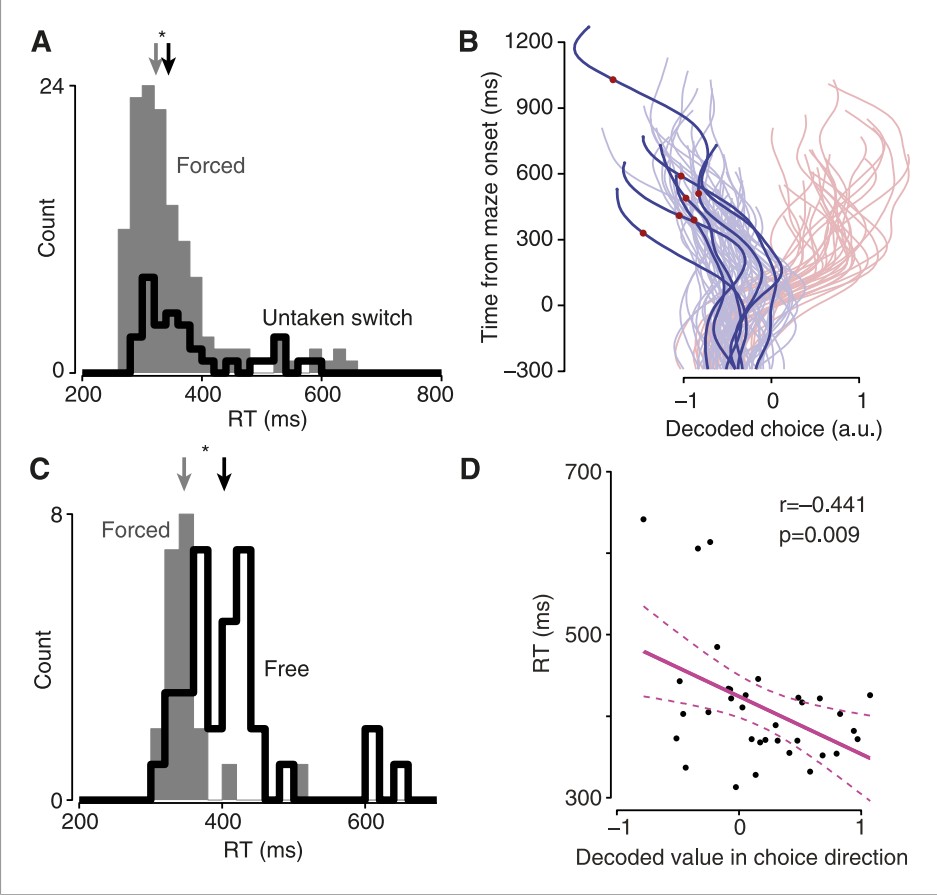

**Figure 5**. Hesitation and indecision. (**A**) RT distribution for forced-choice trials (gray) and encouraged switch trials in which the monkey reached to the always-available target (black). Arrows indicate medians; difference, 21 ms, p = 0.014, Mann–Whitney U test. Dataset J1-T. (**B**) Decoded choice for the untaken switch trials from (**A**) with RTs >450 ms (saturated colors), with forced-choice trials for context (faded colors). Red dots, time of barrier change. (**C**) RT distributions for non-delayed forced (gray) and free (black) trials. Arrows indicate medians; difference, 56 ms, p < 0.001, Mann–Whitney U test. Dataset J2-S. (**D**) Strength of decoded choice in direction of eventual reach (positive toward reach) at 100 ms after maze onset, vs RT for non-delayed free-choice trials. One point per trial; line indicates regression fit, dashed lines show 95% CI of fit.

The following figure supplements are available for figure 5:

**Figure supplement 1**. Decoded choice is undisturbed on slow-RT, untaken-switch trials.

**Figure supplement 2**. Slow-RT free-choice trials were rare in other datasets (J2-S shown in *Figure 5C,D*).

that the decoded choice matched the monkey's intention throughout the delay, since the monkeys did not know the delay duration in advance. Third, we observed many clear changes of mind under conditions where they were expected (free-to-forced, taken-switch, and likely change trials), and much less frequently when they were expected to be uncommon (forced choice, untaken-switch, and unlikely change trials).

While neural responses to forced and free choices were generally similar, the decoder was somewhat less accurate for free choice trials than for forced choice trials. It is not entirely clear why this was the case. One candidate cause of this accuracy difference might have been systematic differences in tuning between forced and free. However, such differences do not seem to be at fault, because decoders trained using free choice trials performed similarly or less well. Other studies have presented two targets that were disambiguated only after the Go cue; in those circumstances, two competing motor plans have been observed both in the saccadic system (*Stanford et al., 2010*;

*Costello et al., 2013*) and for reaching movements (*Cisek and Kalaska, 2005*; *Cisek, 2006*). Here, however, competing motor plans seem unlikely to have interfered with the decoder on most trials: competing motor plans would tend to produce a decoded choice of intermediate magnitude, which was not typical in our data. Instead, we speculate that last-moment changes of mind, occurring after the epoch we decoded, were a more likely culprit. Two pieces of evidence support this hypothesis. First, it is known that responses can change quickly to follow a target that has jumped (*Archambault et al., 2009*, *2011*). Second, our decoder's accuracy was lowest by far in cases where late changes of mind were frequent (taken-switch and likely change trials). This suggests that some of these changes of mind either do not require re-planning (*Ames et al., 2014*) or that re-planning occurs at the very last moment (*Wong et al., 2014*). This ability to change one's mind presumably reflects rich internal processing that can continually reevaluate both what act to make and whether to make it now.

From this work, it is not clear where the decision itself is made. Other studies have argued that PMd may sometimes play a role in decision making (*Pastor-Bernier et al., 2012*; *Thura and Cisek, 2014*). Our data do not bear on this question, and it is possible that the decision machinery for this task may be largely or entirely upstream. However, PMd and M1 apparently reflect the ongoing decision process at a fast timescale (present 'Results' and *Thura et al., 2012*; *Thura and Cisek, 2014*), allowing us to use decoding as a tool for gaining insight into the decision making process.

This ability to decode the motor plan at fast timescales allowed us to examine not only internally-driven changes of mind on single trials, but also features unique to individual datasets or occurring on only a relatively small fraction of trials. For example, indecision (delay in making an initial motor plan when faced with a choice) occurred in only one dataset, yet the decoder could provide insight into how a slowly-evolving initial motor plan might explain the slow RT. Such differences between datasets are likely to occur in rich cognitive behaviors, where individuals' strategies may reasonably differ. On hesitation trials (slowed RT when a new option was presented but not exercised), the decoder revealed that the slow RT was not due to re-planning: no vacillations or weakening of the plan were observed on these trials. This permits comparison with saccadic inhibition (*Reingold and Stampe, 2002*; *Buonocore and McIntosh, 2012*), in which saccades are delayed when a distractor appears shortly after the Go cue. There is evidence that saccadic inhibition may be due to mutual inhibition of competing motor plans (*Bompas and Sumner, 2011*). Hesitation did not obviously involve the formation of a competing motor plan, but it remains possible that competing motor plans are formed in a trigger-related or upstream area, or during other behaviors (such as indecision). As we learn more about each process, a more mechanistic comparison may become possible.

The similarity we observed between activity for forced and free choices suggests that the forced-free distinction may lie largely upstream of the motor cortices. This similarity has the practical consequence that neural prosthetics trained using forced choices will likely generalize well to free choices (*Musallam et al., 2004*; *Santhanam et al., 2006*). In addition, this rapid-choice view challenges previous conclusions, based on trial-averaged data, which argue that humans make decisions unconsciously and only later become aware of them (*Libet et al., 1983*; *Hallett, 2007*). In our data, motor plans were typically formed quickly, corresponding to preliminary choices. However, our data suggest that the moment when neural activity begins to change is not necessarily the moment a decision is finalized (*Schurger et al., 2012*): the movement can be delayed (hesitation) or the choice revised (vacillation).

## Materials and methods

### Subjects

Animal protocols were approved by the Stanford University Institutional Animal Care and Use Committee. Subjects were two adult male macaque monkeys (*Macaca mulatta*), J and N, trained to perform a delayed reach task on a fronto-parallel screen for juice reward. Use of two monkeys is standard practice in the field. The surgical history of the monkeys has been described previously (*Churchland et al., 2010*, *2012*); briefly, they were implanted with a head restraint and two 96-electrode silicon arrays (Blackrock Microsystems, Salt Lake City, UT) in surface M1 and caudal PMd of both monkeys (estimated from local anatomical landmarks). Both monkeys performed the task with their right arm, and the arrays were implanted in the left hemisphere.

## Task details

As described in the 'Results', both monkeys performed a decision-making variant of the center-out delayed-reach 'maze' task. Both had previously performed other variants of the maze task, including thousands of different mazes (*Churchland et al., 2010*; *Kaufman et al., 2013*, *2014*). Typically, they completed novel mazes successfully on the first attempt. Errors for both novel and familiar mazes were almost always due to grazing the corner of a barrier along the correct path; other types of errors were much less common, and reaching toward a blocked target or into a dead end was extremely rare. These patterns of success and failure argue that the monkeys 'understood' the task and were not simply performing memorized stimulus-response associations.

The task was performed with a cursor projected on the screen just above the monkey's fingertip. Initially, the two targets jittered slightly in place, indicating that the monkey was not yet allowed to move. The targets and barriers always appeared simultaneously. 'Go' was indicated by cessation of target jitter, the targets filling in, and the fixation point extinguishing. The monkey was then required to make a brisk movement, curving to avoid the barriers, and end at either target. The delay duration and latency to barrier change were drawn from independent, truncated exponential distributions ($\tau = 500$ ms). For the delay, the distribution was truncated at 1000 ms; for the barrier change, 1200 ms. To keep the task unpredictable, barrier locations and the occurrence and timing of barrier changes were randomized. All starting configurations and possible changes occurred with equal frequency. The two key barriers were not visibly different from the other barriers. Targets yielded equal rewards.

## Identifying overt changes of mind

In previous studies using the maze task, we have excluded reaches whose velocity profiles differed too much from typical (correlation $r < 0.9$). No such attempt was made here, since this could have introduced bias into decision-related analyses. However, to find overt changes of mind, rewarded trials with correlation coefficients less than 0.9 were screened by hand to identify those in which the monkey clearly began moving toward one target then switched to the other. All rewarded trials exhibiting a clear overt change of mind are plotted in *Figure 1E,F* and *Figure 1—figure supplement 1*.

## Neural recordings

Data from these same arrays has appeared in previous publications for both J (*Churchland et al., 2010*, *2012*; *Kaufman et al., 2014*) and N (*Churchland et al., 2012*; *Kaufman et al., 2014*). Since the primary goal was to decode the monkeys' preparatory activity on single trials, all stable single- and multi-unit recordings with at least one spike during the relevant epoch were included (101 units for J1; 132 units for J2; 96 units for N1; 188 units for N2; 196 units for N3). Spike sorting was performed offline using a custom software package (available online as MKsort; https://github.com/ripple-neuro/mksort), and special attention was paid to ensuring the stability of isolations over the session. The total numbers of successful trials per dataset were: J1-T, 1302; J1-S, 1345; J2-T, 1108; J2-S, 1096; N1-S, 1282; N2-S, 869; N3-S, 739. For any given trial subset of interest, the number of trials was necessarily much lower.

## Data selection

Each monkey performed this task for approximately 2 weeks, resulting in eight possible datasets for monkey J and 10 for monkey N. For monkey J, on the first 4 days we did not yet have certain timing equipment in place; one of these datasets was used to pilot analyses but was not included for confirmatory analysis. On 2 days, we changed the mazes mid-session to achieve better behavior; these datasets were not analyzed. The remaining 2 days' data were included. For monkey N, on 6 days we changed the mazes mid-session or the monkey exhibited unstable preferences over the session. Those datasets were not analyzed further. For one dataset, the neural data could not be decoded. In the remaining three datasets, the neural data could be decoded for the S maze but not the T maze. These three S maze datasets were included.

## Similarity of forced- and free-choice responses for single neurons

To determine whether neurons exhibited similar responses for forced and free choices, we performed a correlation analysis. For each neuron, we first collected a vector of trial-averaged firing rates over time for the forced left condition. Firing rates were smoothed with a Gaussian (30 ms SD).

We concatenated this response vector with the smoothed response vector for the forced-right condition. We then correlated it (Pearson correlation) with the analogous response vector for the free-choice conditions. The resulting correlation coefficients were averaged over neurons. This analysis was performed separately for the T-maze and S-maze, and the epoch considered was from −200 to +300 ms from maze onset.

We restricted the above analysis to units with a reasonable signal-to-noise ratio (SNR), since 'modulation' for unresponsive units would simply be noise and produce correlation coefficients near zero. To compute signal, we concatenated the unit's forced and free response vectors then took the range of this vector. To compute the noise, we found the greatest SEM for any point in the concatenated response vector. The SNR was signal divided by noise. We included units with an SNR of at least four. This included a total of 219 units.

## Single-trial decoding

In order to smooth the data in a principled way based on the data itself, and to reduce the possibility of overfitting the classifier, we first applied Gaussian Process Factor Analysis (GPFA; *Yu et al., 2009*). Among other advantages, GPFA chooses smoothing kernels for each dimension to reflect both their natural spectral content and the signal-to-noise available in the data. For each trial, we took the portion of the data from 300 ms before maze onset to 200 ms before movement onset, and applied GPFA with a 20 ms bin width and 12 dimensions. Only successful trials were included. The dimensionality was chosen conservatively based both on previous analysis of PMd activity and cross-validation of the present data, which indicated that any dimensionality between ∼8–14 would retain most structure without increasing noise. Movement epoch data were excluded because firing rate changes tended to be much larger and thus dominated the resulting space, reducing the quality of the delay epoch trajectory estimation. For both monkeys, GPFA was performed on all successful trials from a single day together. The subsequent decoding did not appreciably change quality when performing the GPFA step on the T-maze and S-maze separately.

As our classifier, we trained a Support Vector Machine (SVM; *Cortes and Vapnik, 1995*) with a linear kernel (using the svmtrain function in Matlab, Mathworks, Natick, MA). The linear SVM identifies a classifying hyperplane (in this case, in the 12-D GPFA space) based on a linear weighted sum of the dimensions. Since the GPFA step is linear as well, the resulting classifier is simply a weighted sum of the neurons' smoothed spike trains.

We chose to use an SVM for two reasons. First, the SVM is a wide-margin classifier, which is both inherently regularizing and well suited to non-Gaussian data. Second, in practice it performed qualitatively and quantitatively better in cross-validation than other decoders tested (such as logistic regression or a Naïve Bayes classifier). To obtain a graded decode of choice, we projected the data onto the vector normal to the hyperplane (*Figure 2C*). This is equivalent to taking the signed distance from the classifying hyperplane.

We trained the SVM using only forced-choice trials with delay durations of at least 300 ms. We used only these relatively-long-delay trials so that early delay activity was not overrepresented. To ensure that the decoder was trained on activity that at least partially reflected a choice, only time points from 80 ms after maze onset to 200 ms before movement onset were considered from these trials. For convenient scaling, the resulting decoded values were normalized by the 90th percentile of all decoded values for that dataset (not just forced-choice trials).

In order to test the decoder's performance with time (*Figure 3A*), we divided the distribution of delay durations into 60 ms bins. For each dataset, we then identified all the trials for which the last decoded time point happened to fall in that bin. The value reported is the fraction of trials for which the final point was decoded correctly; that is, whether it agreed with the monkey's subsequent choice.

We computed a p-value to verify that the cross-validated forced-choice decoder performed above chance. To do so, we performed a simulation in which we chose the 'decoded choice' for each trial to be left or right by chance (according to their overall prevalence), then checked how often this simulation performed as well as the real decoder. 10 million runs were performed with different random seeds.

## Trial selection

For all decoder-based analyses except those noted, only trials with delay durations ≥300 ms were used. Where noted, trials with delay durations of zero were used (the Go cue was presented simultaneously with maze onset). For producing PSTHs, trials with a delay duration of ≥350 ms were

used. When forced or free choice trials are referenced without explicitly noting barrier changes, only trials with no barrier changes were included.

### Identifying possible changes of mind for PSTHs

For producing PSTHs, the presence of changes of mind represented a potential confound. We therefore used the moment-by-moment decoded choice to aggressively exclude all trials that might have contained such changes of mind. Specifically, we excluded all trials for which the eventual choice did not match the decoded choice at any point more than 160 ms after maze onset (150 ms is the approximate length of time it takes for movement preparation to complete, see *Churchland et al., 2006b*; we rounded this value up to the next multiple of 20 ms to align with our binning). This criterion was applied to both forced choice and free choice trials. Change-of-mind trials were also excluded before determining whether selectivity was consistent for forced and free choices (exclusion for free choices only). This exclusion was not performed before any other analyses.

### Quantifying changes of mind in free choice, encouraged switch, and biasing change trials

To identify trials with clear covert changes of mind, we used the following algorithm. For each trial with a delay of ≥300 ms, we considered time points ≥160 ms after maze onset (see above for justification of timing). After this time, to be considered a change of mind the time course of the decoded choice had to meet three criteria: (1) at some time point, it needed to change from leftward to rightward or vice versa; (2) at some time point, it needed to be ≥10 times as likely to come from the 'leftward' forced-choice distribution of decoded choice as from the 'rightward' forced-choice distribution of decoded choice for the same time point; and (3) at some time point, it needed to be ≥10 times as likely to come from the 'rightward' forced-choice distribution of decoded choice as from the 'leftward' forced-choice distribution of decoded choice for the same time point. That is, the decoded choice needed to swing from 'strongly left' to 'strongly right' or vice versa. To determine likelihoods, Gaussian fits were used for the leftward and rightward distributions. For determining the forced-choice distributions of decoded choice at late time points, all time points ≥600 ms were pooled. To be conservative, when this algorithm was applied to leave-one-out cross-validation trials, the classifier and distributions from that cross-validation fold were used. All trials meeting these criteria are shown in the relevant plots. This algorithm was also used to identify trials for *Figure 4G,H*.

### Determining whether changes of mind passed near the baseline state

We determined whether the neural state during a change of mind resembled the baseline state, in the 12-D GPFA space (*Figure 4G,H*, *Figure 4—figure supplement 5*). This 12-D state acts as a denoised summary of the firing rates of all the recorded neurons.

We first determined the mean baseline neural state. To do so, we averaged the neural states of all successful trials from −300 to −40 ms from maze onset.

We then found the distribution of the single-trial moment-by-moment baseline states around the mean. To do so, for each of the points from the previous step, we computed the Euclidean distance between the point and the mean baseline state. This distribution across trials and times is shown in gray.

Next, we identified change of mind trials: subsets of free choice trials and biasing change trials (which are free choices throughout, but in which one barrier changes difficulty during the delay). In each case, we identified these trials using the algorithm described in the section above. We then found the 'crossings', the time points just before and just after the decoded choice changed sign. Each trial thus contributed two points. Finally, for each of these points, we found the Euclidean distance to the mean baseline state as above. The resulting distributions are plotted in filled pink and green (*Figure 4G,H*, *Figure 4—figure supplement 5*).

Finally, for comparison, we considered all free choice trials and all biasing change trials (not just the crossing trials). For each point at least 160 ms after maze onset, we again found the Euclidean distance to the mean baseline state. These distributions are plotted in hollow purple and green.

### Determining whether changes of mind had typical firing rates

To compare firing rates during changes of mind with firing rates on other trials, we had to choose a comparison population of trials and a matched window of time (since firing rates were not perfectly

stable over the delay). We therefore used the following procedure. For each unit, for each vacillation trial, we found the last time that the decoder switched from indicating one choice to the other. We then found the firing rate of that unit in the 200 ms window centered on this crossing time. Next, we found the mean firing rate in the same window for the same unit for all non-vacillation free left-choice trials, and for all non-vacillation free right-choice trials. After repeating this procedure for each vacillation trial, we found three average firing rates: for all the vacillation trials, for all non-vacillation free lefts, and for all non-vacillation free rights. Finally, we categorized the vacillation firing rate as lower than both free rates, between them, or higher than both free rates. The procedure for bias trials was similar: analogously, we compared bias trials on which the decoder did indicate a change of mind (sign change in the decoded choice) with those in which it did not. If instead we repeated the analysis using free choice non-vacillation trials for comparison, results were similar. This last point indicates that the observed difference between vacillations and induced changes of mind were not due to differences in the comparison trials.

## Regression-based indecision analysis

We wished to see whether a slowly resolving decoded choice might be responsible for the slow RTs sometimes observed on free choice trials (*Figure 5D*, *Figure 5—figure supplement 2*). To do so, we first identified free choice trials on which the Go cue was presented simultaneously with maze onset, and which evoked an RT $\geq$300 ms. For these trials, we considered the decoded choice at 100 ms after maze onset. For the regression, we ensured that a decoded choice agreeing with the eventual choice was positive while a decoded choice disagreeing with the eventual choice was negative. Thus, for trials on which the monkey chose the leftward target, we inverted the sign of the decoded choice. Linear regression was then performed.

## Acknowledgements

We thank M Mazariegos for expert surgical assistance and veterinary care; R Kiani, C Chandrasekaran, D Peixoto, and W Newsome for scientific discussion, D Haven and B Oskotsky for technical support, and S Eisensee, E Castaneda, and B Davis for administrative support. This work was supported by a National Science Foundation graduate research fellowship (MTK), Swartz Foundation fellowship (MTK), Burroughs Wellcome Fund Career Awards in the Biomedical Sciences (MMC, KVS), the Christopher and Dana Reeve Foundation (SIR, KVS), NIH CRCNS R01-NS054283 (KVS), an NIH Director's Pioneer Award 8DP1HD075623 (KVS), and DARPA REPAIR N66001-10-C-2010 (KVS).

## Additional information

### Funding

| Funder | Grant reference | Author |
| --- | --- | --- |
| National Science Foundation (NSF) | Graduate Student Fellowship | Matthew T Kaufman |
| Burroughs Wellcome Fund | Career Awards in the Biomedical Sciences | Mark M Churchland, Krishna V Shenoy |
| Christopher and Dana Reeve Foundation | | Stephen I Ryu, Krishna V Shenoy |
| National Institutes of Health (NIH) | CRCNS R01-NS054283 | Krishna V Shenoy |
| National Institutes of Health (NIH) | Director's Pioneer Award 8DP1HD075623 | Krishna V Shenoy |
| Defense Advanced Research Projects Agency (DARPA) | REPAIR N66001-10-C-2010 | Krishna V Shenoy |
| Swartz Foundation | Postdoctoral fellowship | Matthew T Kaufman |

The funders had no role in study design, data collection and interpretation, or the decision to submit the work for publication.

## Author contributions

MTK, Conception and design, Acquisition of data, Analysis and interpretation of data, Drafting or revising the article; MMC, Conception and design, Acquisition of data, Drafting or revising the article; SIR, Performed array implantation surgery; KVS, Conception and design, Analysis and interpretation of data, Drafting or revising the article

## Ethics

Animal experimentation: This study was performed in strict accordance with the recommendations in the Guide for the Care and Use of Laboratory Animals of the National Institutes of Health. All of the animals were handled according to approved institutional animal care and use committee (IACUC) protocols of Stanford University. All surgery was performed under isofluorane anesthesia with complementary analgesia, and animals were closely monitored for distress. The ethics protocol number will be made available to scientific or funding groups upon request to the authors.

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
