## [Decision Letter]

Thank you for sending your work entitled “Moment by moment: Vacillation, indecision and hesitation in monkey motor cortex” for consideration at *eLife*. Your article has been evaluated by Eve Marder (Senior editor) and three reviewers, plus a member of our Board of Reviewing Editors. Their opinion is favorable provided that you can address two main concerns described below, which entail additional analysis of behavioral data.

This manuscript describes population activity in premotor and motor cortex during single trials to study internal events occurring during free choice, such as changes of mind and vacillations. The results suggest that initial motor plans, even those that stem from free choices, are sometimes corrected or changed on the fly. Previous studies have reported evidence for such ‘vacillations’ before, but here the aim is to describe them in single trials. The results are likely to be of interest to a wide range of neuroscientists studying decision making and motor control. However, there are two main concerns.

First, the data are not very abundant, particularly for one of the monkeys. Only 5 datasets (∼3 days of recordings) had sufficient statistical power for single trial analysis. This leads to multiple questions. Five out of how many? What happens in the rest of the datasets? Does the decoder not work for them? How many neurons are in each data set? What is the sensitivity of the decoder to the number of neurons? How homogeneous is the weight of each neuron in the decoder?

The paucity of data in turn reflects on the other main concern, namely whether both monkeys indeed engaged in “decisions” and occasional “free choices”. In one monkey, indeed, the supposed free choice conditions led to lopsided behavior (dataset J2, Table 1): for the T-maze, the animal moved left 92% of the time when that choice was easy, and 0% when it was hard (“diff”). Similarly lopsided percentages were seen for the S-maze, where the animal moved to the left nearly 100% of the time when the rightward choice was hard, and 6-23% of the time when it was easy.

Though these data may reflect truly free choices, a more prosaic explanation is that they reflect a learned, arbitrary sensorimotor association between a screen configuration and a reach movement. In the rare cases when the animal makes a movement to the other target (e.g. 8% of the time to the right in the easy|easy condition for J2T), it may simply mean that he hasn't learned the association well enough and is making an “error” with regard to the association. An incomplete learning of the association may also help explain the activity of the other monkey, where the percentages are less lopsided.

If the monkey has learned to make a movement to the left when it sees a particular stimulus configuration, then it is hard to argue that it is making a free choice. For example, drivers learn to stop at a red light. But is that a decision in any interesting sense, i.e. does it involve weighing the outcomes of different alternatives and choosing one based on one's values or preferences, or after accumulating enough evidence about the correct answer? It is thus possible that the data here reflect arbitrary sensorimotor associations, which have been studied for years, particularly in PMd. In such learned associations between a stimulus and response, activity during the delay period is simply preparation for translating a sensory stimulus to a particular upcoming movement.

These two concerns are related: perhaps by analyzing more behavioral data (concern 1) it is possible to help the case for free choices (concern 2). Indeed, the current Figure 2 lumps together all the conditions, giving the potentially misleading impression that the animal is “freely” choosing to go either to the left or the right with nearly equal frequency. It would be useful to analyze more behavioral data, and to provide additional analyses. One could be a summary bar plot comparing the percentage of vacillations across all conditions for a given data set, i.e., forced choices, free choices, taken/untaken switches and likely/unlikely changes, indicating whether those percentages differed significantly across conditions (say, based simply on their overall occurrence and binomial statistics).

Other issues:

Why do all the decoded values start with a leftward (negative) bias (for instance, Figure 2, x axis)? Because crossing the zero value is a criterion for determining a vacillation, it seems important to ensure that decoded values computed before the maze onset are not artificially biased, as this could potentially affect the labeling of trajectories as vacillations.

The monkeys show highly uneven preferences for each target (Table 1). Is there a neural correlate of this bias?

The present analysis makes no effort to separate neurons recorded from M1 and PMd. Yet, this group has recently documented a difference in preparatory activity between these two areas. Why is such a pooling acceptable? It would be more interesting to see a comparison between the two areas. What happens if the analysis is done for each area separately?

The vacillations in the decoder in the free choice condition (Figure 2) look extremely stereotyped further supporting the idea that this condition is not reflecting a free choice but rather the temporal dynamics of planning of an arbitrary sensorimotor association. In both panels, the leftward movements almost always shift to the right first before shifting to the left.

It seems odd that vacillations are not associated with longer reaction times (RT). There was an effort to see if long RTs were associated with vacillations, but the reverse selection criterion might be more fruitful: having identified single trials as vacillations, are their RTs longer?

The manuscript states (in the Results section) that neuronal responses encoded forced and free choices similarly, and emphasizes that this result was not necessarily expected. It should be quantified more rigorously. How was the similarity between neuronal responses assessed? What were the reaction times for the respective choices? Using tuning preference as the only measure to make this point is rather weak.

The manuscript argues that because the time course of the neural response is nearly identical for forced and free trials, the monkey made up his mind quickly when presented with a free choice. This claim is unsubstantiated. An alternative explanation may be that the monkey rapidly integrated sensory information and associated a particular stimulus configuration with a particular movement.

Also in the Results, the authors state: “more traces cross the midline for free choices than forced choices.” This should be quantified.

Still in the same section: “on roughly half of such trials, the monkey should have already fortuitously chosen the ‘lucky’ side.” Given that choice probabilities were not 50/50, what proportion of the time did the monkey actually choose the “lucky” side?

The following sentence, in the Results: “in both monkeys, a modest number of trials exhibited spontaneous vacillations.” Since decoded choices are ultimately a reflection of population firing rates, is there some feature of the population firing rate that was different on vacillated trials? Namely, were firing rates lower? If overall population firing rates were similar between vacillated trials and regular trials, is there a subset of neurons that changes firing rate? It would be good to understand if/how neural firing rates in both single cells and the population give rise to vacillations.

In the Results, the authors state: “to our knowledge, observations of such spontaneous vacillations when presented with a free choice—without changing external evidence—have not previously been described”. This claim appears unsubstantiated. Vacillations could simply be measurement error in the neural trajectory. Indeed, in 40% of the datasets, there was no significant difference in the number of vacillated trials between free and forced choice conditions.

Discussion, first paragraph: Consider comparing these results with the work of Stanford and colleagues on saccades (Nat Neurosci 13:379, 2010; J Neurosci 33:16394, 2013), which also demonstrates competition between motor plans within the same task, from quick choices to slower trials in which the initially dominant motor plan is suppressed and overtaken by the alternative one (’changes of mind'). Although that work is based on trial-averaged data, it tightly relates those motor planning patterns to behavior, and to RT in particular.

Discussion, second paragraph: “our data seem at odds with standard integrate-to-bound models.” In what way are they incompatible? Most of the neurons in Figure 2—figure supplement 1 differentiate one choice from the other in less than 200 ms relative to the onset of the maze, showing an initial rise and differentiation that resemble the data from saccade-based tasks. (Responses are much more variable when aligned on movement onset, but that is understandable given the dependencies of PMd and M1 neurons on kinematic parameters). Second, such rise-to-threshold models have been used successfully in a variety of choice tasks, not just in random-dots tasks ’when information is delivered slowly.‘ But if ‘ramping’ is understood simply as an increase in motor planning activity, then the classic ramping associated with saccades is indeed very quick, on the order of 200 ms (Science 274:427, 1996; Neuron 76:616, 2012). Again, the work of Stanford and colleagues illustrates that a scheme in which alternative motor plans compete to reach a threshold works well even when decisions are urgent and choices quick. The work of Cisek and colleagues is consistent with such competition too, but over much longer time scales (Neuron 45:801, 2005; J Neurosci 29:11560, 2009; Neuron 81:1401, 2014). Thus, the rate at which sensory information is delivered or perceived is orthogonal to this issue of how to characterize motor-planning activity.

The following statement in the Results: “the slow RTs on these trials were not due to re-planning, but instead involved a hesitation to execute.” The high RTs on untaken switch trials (Figure 4) is reminiscent of a phenomenon known as saccadic inhibition, in which a the appearance of an irrelevant distracter slows down a planned saccade to an unambiguous, unique, predictable target (J Cogn Neurosci 14:371, 2002; J Neurosci 31:12501, 2011; Vision Res 69:32, 2012). This is as if a sensory event (a change in a barrier, in this case) transiently interrupted and delayed an ongoing motor plan.

Figure 2: what are the dots on the decoder trajectories in this case?

Free-to-forced trials: it may be nice to show the decoder in these trials before and after the change in the maze.

Forced-to-free trials: how do we explain trials without a detectable change of mind?

What happens during unsuccessful trials?

The title is compelling, but the three phenomena it lists—vacillation, indecision and hesitation—are not clearly distinguished in the manuscript. It would be helpful if the manuscript described them better.

For instance, the definition of a vacillation (or change of mind, but better stick to one word) is clearly explained in the Methods, but it would be useful if the main text mentioned that they were identified as big swings in the decoded choice.

Results, tenth paragraph: consider adding some clarification when mentioning “…on free-choice trials…”. Rather, the lower performance may reflect somewhat higher variability or complexity during free choices or something similar.

---

## [Author Response]

*[…] The results are likely to be of interest to a wide range of neuroscientists studying decision making and motor control. However, there are two main concerns*.

*First, the data are not very abundant, particularly for one of the monkeys. Only 5 datasets (∼3 days of recordings) had sufficient statistical power for single trial analysis*. *This leads to multiple questions. Five out of how many? What happens in the rest of the datasets? Does the decoder not work for them?*

We understand the concern about the small number of days’ data included. First, we would like to note that the 5 datasets originally included constitute a large amount of data: we recorded many trials per day on a complex task while performing many simultaneous high-density neural recordings, resulting in >700,000 unit-trials.

Nonetheless, to address this concern we screened 3 additional days’ data for monkey N (this had not been done before the reviewers’ request). Two produced sufficiently good decoders, and are included in the revised manuscript. The “new” datasets have lower trial counts, which is why they were not analyzed previous to this request. Nonetheless, they resulted in adequate decoding. We thus now have 7 datasets instead of 5 (40% more data), spanning 5 days of recording. The analysis of each dataset in detail now occupies 16 figures and 3 tables, including 150 figure panels.

Concerns about data selection are understandable. To address this concern, we have added a section to the Methods which summarizes all 18 days’ data collected and how we chose the five days (seven datasets) included in the paper (please see the subsection “Data selection” in Materials and methods.)

How many neurons are in each data set?

Ranges are now included in the Results in addition to the Methods. The datasets included 101, 132, 96, 188, 196 single- and multi-units (datasets J1, J2, N1, N2, N3).

What is the sensitivity of the decoder to the number of neurons?

In general, this is an interesting question when developing neural decoders (for example, for neural prosthetics). In the present work the decoder is used as a tool, and is not itself intended as a focus; nonetheless, we agree that it is important to perform some basic checks on the decoder as the reviewers suggest. We describe two such checks below.

The decoder quality does not seem to strongly track the number of units across datasets. In monkey N, free choice decoding was most accurate for the dataset with the smallest unit count, while forced-choice decoding was better for the datasets with higher units counts. The reasons for this may be complex. First, trial count (and not just unit count) is important in identifying the optimal GPFA space and decoder. Second, GPFA automatically selects the smoothing kernels for each dimension, and when the signal-to-noise is lower it can partly compensate by choosing to smooth over a little more time. We now note this in the Methods (in the subsection headed “Single-trial decoding”): “Among other advantages, GPFA chooses smoothing kernels for each dimension to reflect both their natural spectral content and the signal-to-noise available in the data.”

Nevertheless, we wished to test for sensitivity to unit count within each dataset. We followed the reviewers’ suggestion of decoding from PMd and M1 separately (see below for more details). This cuts unit counts roughly in half. We found that we could readily decode choice from the two areas separately, implying that information is distributed across many units and both areas, and that decoder performance tends to decline slowly with unit count. That said, the decoder performance is of course somewhat less good when using only half the data. The values are given in the new Tables 2 and 3.

How homogeneous is the weight of each neuron in the decoder?

The weights tended to fall off smoothly, approximately as a double exponential or power-law distribution (for a typical dataset, J1–T, see Figure 6). The one possible exception was dataset N1, in which one unit was strongly overrepresented (coefficient 80% larger than the next unit). The distributions were also smooth if we multiplied each unit’s coefficient by the standard deviation of that unit’s firing rate to get an estimate of the “influence” of the unit (Figure 6, second row). We also examined the decoders’ weights relative to the GPFA dimensions. Here again the distributions mostly fell off smoothly (Figure 6, bottom row).

Author response image 1.This figure shows how the decoder depends on the units (top two rows) and the GPFA dimensions (bottom row) for dataset J1-T. We allowed a bias term for the power law fits; without the bias term, the double-exponential was clearly the better fit, and thus is the distribution mentioned in the manuscript.**DOI:**
http://dx.doi.org/10.7554/eLife.04677.027

After looking carefully at these results for each dataset, our overall conclusion was that the decoders relied on many units. We have summarized this conclusion in the Results (in the subsection headed “Decoding the moment-by-moment population response”): “This decoder was simply a weighted sum of the neurons’ smoothed spike counts (Figure 2). Empirically, the decoder chose to exploit the activity of most neurons, with the decoder’s weights distributed over units approximately as a double-exponential distribution (not shown).”

*The paucity of data in turn reflects on the other main concern, namely whether both monkeys indeed engaged in “decisions” and occasional “free choices”. In one monkey, indeed, the supposed free choice conditions led to lopsided behavior (dataset J2, Table 1): for the T-maze, the animal moved left 92% of the time when that choice was easy, and 0% when it was hard (“diff”). Similarly lopsided percentages were seen for the S-maze, where the animal moved to the left nearly 100% of the time when the rightward choice was hard, and 6–23% of the time when it was easy*.

*Though these data may reflect truly free choices, a more prosaic explanation is that they reflect a learned, arbitrary sensorimotor association between a screen configuration and a reach movement. In the rare cases when the animal makes a movement to the other target (e.g. 8% of the time to the right in the easy|easy condition for J2T), it may simply mean that he hasn't learned the association well enough and is making an “error” with regard to the association. An incomplete learning of the association may also help explain the activity of the other monkey, where the percentages are less lopsided*.

*If the monkey has learned to make a movement to the left when it sees a particular stimulus configuration, then it is hard to argue that it is making a free choice. For example, drivers learn to stop at a red light. But is that a decision in any interesting sense, i.e. does it involve weighing the outcomes of different alternatives and choosing one based on one's values or preferences, or after accumulating enough evidence about the correct answer? It is thus possible that the data here reflect arbitrary sensorimotor associations, which have been studied for years, particularly in PMd. In such learned associations between a stimulus and response, activity during the delay period is simply preparation for translating a sensory stimulus to a particular upcoming movement*.

*These two concerns are related: perhaps by analyzing more behavioral data (concern 1) it is possible to help the case for free choices (concern 2). Indeed, the current*
Figure 2
*lumps together all the conditions, giving the potentially misleading impression that the animal is “freely” choosing to go either to the left or the right with nearly equal frequency. It would be useful to analyze more behavioral data, and to provide additional analyses. One could be a summary bar plot comparing the percentage of vacillations across all conditions for a given data set, i.e., forced choices, free choices, taken/untaken switches and likely/unlikely changes, indicating whether those percentages differed significantly across conditions (say, based simply on their overall occurrence and binomial statistics)*.

Thank you for raising this important question, which we certainly wish to address squarely and have it come across clearly in the manuscript. We have now added discussion of choice in our task, a new behavioral analysis, and the neural analysis suggested to address this point. These are described in turn.

First, we have added a brief discussion of both monkeys’ previous experience in the task description, which argues that they understand the options well (in the subsection headed “Task details”):

“As described in the Results, both monkeys performed a decision-making variant […] were not simply performing memorized stimulus-response associations.”

Second, one gold standard for decision making is the presence of overt changes of mind (e.g., Resulaj et al. Nature 2009). These were present in our data, and we have now plotted them (Figure 1; Figure 1—figure supplement 1). As we now note in the paper (in the third paragraph of the Results section):

“In most of these overt change-of-mind trials, both options were available when the monkey deviated from his original reach. Depending on the trial, these changes of mind were driven either by a barrier change that only altered the relative difficulties of the two options, or by some purely internal process.”

In sum, (1) the monkeys seem to “understand” the options (given their prior experience and the near-total absence of inappropriate reaches), (2) the same stimulus elicited two different behaviors on different occasions, (3) small changes in the stimulus pushed around the probabilities of these behaviors in the expected ways, and (4) we observed overt changes-of-mind. This argues that the monkeys really did make choices about which target to reach to. This logic is now better outlined in the paper (in the third paragraph of the Results section):

“In agreement with this idea, the monkeys’ choices statistically reflected the relative difficulties of the two options (Table 1) […] they began reaching toward one target but performed a sharp turn and reached to the other target (Figure 1, Figure 1—figure supplement 1).”

Regarding biases, our task is quite different from many others such as the random dots task or the tokens task. In those tasks, there is a zero-coherence or balanced-information condition, and the monkey must guess at the “true” answer. Our task instead presents truly free choices. Either alternative will get the monkey the same reward, as long as he is fast and accurate. Thus, it is both expected and inevitable that the monkeys will have some biases. Critically, though, for all the analyzed datasets we succeeded in creating conditions where choices were in fact distributed between two options. This did not always occur for all combinations (e.g., it might be true for easy|hard but not hard|hard) and the distribution was rarely 50/50, but that is expected from the nature of the task. In each dataset we had at least two (and up to four) cases where choices were distributed among both options.

To explain the nature of the task and the leeway for bias, we were inspired by the reviewers’ red light analogy. We now include, in the Results section, a traffic analogy that fits more closely:

“This task presented a continuum of situations, ranging from complete experimenter control (“forced choices,” with only one target accessible), to moderate experimenter influence […] Our task is roughly analogous: often the stimulus does not fully specify the right answer, leaving a choice to be made.”

“While biases were common (much as they would be for drivers choosing lanes) both monkeys made reaches to both targets when given a choice, and in cases where one target was easier, it was chosen more often.”

To address the concern about confusion over what is plotted in Figure 2, we have added a clarification (in the subsection headed “Free choices”):

“Neural data from all the long-delay free choice trials (including four barrier configurations), in which neither target was blocked and in which no barrier changes occurred, are shown in Figure 2 (saturated traces, forced-choice trials shown as faded traces for context; see also Figure 2—figure supplement 2; p<10^-6^ for all datasets).” The probabilities of each choice under different circumstances are included in Table 1.

Finally, we have added the neural analysis suggested: a bar plot comparing how often the decoder exhibits large swings under different circumstances (Figure 4). This makes it clear that we see these swings much more often when we expect them to be common than when we expect them to be rare. The associated statistics are also very strong. We found the plot surprisingly satisfying, and thank the reviewers for suggesting we make it.

*Other issues*:

*Why do all the decoded values start with a leftward (negative) bias (for instance,*
Figure 2*, x axis)? Because crossing the zero value is a criterion for determining a vacillation, it seems important to ensure that decoded values computed before the maze onset are not artificially biased, as this could potentially affect the labeling of trajectories as vacillations*.

This is an interesting point. There are two questions here: (1) Why is the baseline state decoded as “left”? and (2) Does this affect our labeling of trials as vacillators?

Considering the second question first, we have added the following control (in the subsection “Spontaneous changes of mind (vacillations) are present in the neural activity”:

“Second, we addressed the concern that some of these vacillations might be related to the tendency for the baseline state (before maze onset) to produce a “leftward” decoded choice in most datasets. To check for this possibility, we performed a simple control. For each vacillation trial, we considered the last time the decoded choice switched sign. We counted how many times it was left-to-right, and how many times it was right-to-left. In 3/7 datasets, left-to-right switches were more common; in 3/7, right-to-left were more common; in 1 dataset they were tied.”

Regarding the first question, it is not entirely clear why the baseline state tended to be decoded as “left”. It was possible to find decoders that were more symmetrical, but the best-performing decoders we could identify were typically asymmetric. This may be related to the fact that rightwards reaches were contralateral to the recording arrays, but one can readily find neurons that increase their rates for leftward reaches instead of rightward. We have added the following text to the manuscript to explicitly note this point (at the end of the subsection “Spontaneous changes of mind (vacillations) are present in the neural activity”): “It is not entirely clear why baseline decodes tended to register as ‘leftward’, but it does not appear that baseline bias affected our ability to identify vacillations.”

*The monkeys show highly uneven preferences for each target (*Table 1*). Is there a neural correlate of this bias?*

There are two possible neural correlates of the bias. First, and most straightforwardly, one expects to see more ‘neural choices’ in the biased direction. This was very much the case. Second, there might also be a bias during the pre-maze period. We did not find such a pre-existing bias.

Regarding the lack of a pre-existing bias, this is likely because the two maze types (T and S) were randomly interleaved on a trial-by-trial basis for both monkeys, so neither monkey could anticipate what options he would have on the next trial. Thus, we don’t expect to see preparation before maze onset. As can be seen in the pre-maze-onset portion of the free-choice decodes (Figure 2), baseline activity is not predictive of the monkey’s choice. Moreover, attempts to decode the monkey’s future choice from baseline activity did not succeed. This is now noted (in the subsection headed “Free choices”): “As expected, decoding was unsuccessful before maze onset, for either forced or free choices.”

The present analysis makes no effort to separate neurons recorded from M1 and PMd. Yet, this group has recently documented a difference in preparatory activity between these two areas. Why is such a pooling acceptable? It would be more interesting to see a comparison between the two areas. What happens if the analysis is done for each area separately?

This is an important question: do the activity patterns for PMd and M1 agree? Answering this question is a bit technically challenging however, as it requires us to split our neural data in half. This is made still more difficult because our M1 recordings happened to have fewer units than PMd, and M1 generally has less preparatory activity. Despite the resulting signal-to-noise challenge, the two areas showed remarkable agreement. We have added two new tables showing the decoder performance for PMd and M1 separately and the fraction of time the separate decoders agreed, assessed over all trials (Table 2) and free choice trials only (Table 3). In addition, we have added a figure showing a few example vacillation trials in which the two areas exhibited the same vacillation (and one trial where they did not). The new accompanying text reads:

“We performed two additional controls to aid in interpreting these apparent vacillations. […] we could often see similar vacillations when decoding PMd and M1 separately (Figure 4—figure supplement 2), though this was not always the case (Figure 4—figure supplement 2).”

*The vacillations in the decoder in the free choice condition (*Figure 2*) look extremely stereotyped further supporting the idea that this condition is not reflecting a free choice but rather the temporal dynamics of planning of an arbitrary sensorimotor association. In both panels, the leftward movements almost always shift to the right first before shifting to the left*.

We agree with the reviewers that there is one dataset for which vacillations were somewhat stereotyped (J2-S; Figures 2 and 4). This stereotypy likely reflects exactly the momentary “indecision” that we found in this one dataset (though we could not measure indecision in long-delay trials, since it has resolved by the time of the Go cue; see next reply). This stereotypy was not present in any of the other six datasets (see Figure 4 and Figure 4—figure supplement 1), and, as expected, neither was indecision.

*It seems odd that vacillations are not associated with longer reaction times (RT). There was an effort to see if long RTs were associated with vacillations*, *but the reverse selection criterion might be more fruitful: having identified single trials as vacillations, are their RTs longer?*

This question gets at a fundamental technical challenge in examining vacillations. For trials in which we can observe neural vacillation, the vacillation must be complete (or nearly so) by the time the Go cue comes. Otherwise, since we don’t analyze peri-movement data, we wouldn’t have seen the vacillation. Since the vacillation is finished by the time of the go cue, we don’t expect those trials to have altered RTs.

That is why we posed the question as we did. In one dataset, free choice trials without a delay often evoked slow RTs. We had the same intuition as the reviewers: perhaps this slowing was due to vacillation. We found that, for these trials, an “incorrect” plan (one that disagrees with the monkey’s eventual reach) 100 ms after maze onset indicated indecision and therefore predicted a slow RT (Figure 5).

*The manuscript states (in the Results section) that neuronal responses encoded forced and free choices similarly, and emphasizes that this result was not necessarily expected. It should be quantified more rigorously. How was the similarity between neuronal responses assessed? What were the reaction times for the respective choices? Using tuning preference as the only measure to make this point is rather weak*.

We agree that similarity of forced and free responses needed to be assessed more thoroughly. We have now performed and included a new correlation analysis as a stronger and more quantitative test. We have added the following text:

In Results (in the subsection headed “Multi-unit responses during forced and free choices”):

“… neurons’ trial-averaged responses during the delay strongly correlated for forced and free choices (mean r=0.76, p<0.001 for 6/7 datasets, p<0.01 for N3-S; sign test; Materials and methods).”

In Materials and methods:

“Similarity of forced- and free-choice responses for single neurons:

To determine whether neurons exhibited similar responses for forced and free choices, we performed a correlation analysis. […] We included units with an SNR of at least four. This included a total of 219 units.”

Regarding reaction times, RTs were similar to both targets for all conditions. This is especially important for the result shown in Figure 5, and we now mention it (last paragraph of the Results section): “These indecisive moments were present for trials resulting in either left or right choices (RT distributions were generally similar for left and right).”

*The manuscript argues that because the time course of the neural response is nearly identical for forced and free trials, the monkey made up his mind quickly when presented with a free choice. This claim is unsubstantiated. An alternative explanation may be that the monkey rapidly integrated sensory information and associated a particular stimulus configuration with a particular movement*.

This section (“Multi-unit responses during forced and free choices”) has been rewritten to be more neutral. It now reads:

“Interestingly, for most neurons, the time course of the neural response was nearly identical for forced and free choices. This implies that the monkey formed a motor plan quickly when presented with a free choice: that is, the time to make an initial movement selection for free choices did not take appreciably longer than the time to determine which target was accessible for forced choices.”

Also in the Results, the authors state: “more traces cross the midline for free choices than forced choices.” This should be quantified.

This section has been rewritten to make clearer that it is an initial intuition-building and that the quantification comes later (in the last paragraph of the subsection “Free choices”):

“It is immediately clear from these free-choice plots that there are some trials in which the decoder first indicated a plan toward one target […] then quantify these changes of mind.”

Several additional statistics have also been added regarding the relative rates at which free and forced choices exhibit large changes in the decoded choice, and these statistics have been similarly expanded for taken and untaken switch trials and likely and unlikely change trials. Summaries for each are also now included in a new plot suggested by the reviewers, Figure 4.

*Still in the same section: “on roughly half of such trials, the monkey should have already fortuitously chosen the ‘lucky’ side*.*” Given that choice probabilities were not 50/50, what proportion of the time did the monkey actually choose the “lucky” side?*

The fraction of time that the monkey will have chosen the “lucky” side is independent of his choice probabilities because of how we chose which side would have a barrier change. Imagine, for example, that for a particular barrier configuration the monkey always plans to the left. If half the time we block the left side, and half the time we block the right side, he will still be “lucky” half the time despite his bias. This is true for any monkey’s choice probabilities as long as we choose which side to block with equal frequency.

We thank the reviewers for pointing out that readers are likely to stumble here. We have now emphasized this reasoning in the relevant sentence: “Since we randomly chose which side to make inaccessible, on roughly half of such trials the monkey should have already fortuitously chosen the ‘lucky’ (unchanging) side.”

*The following sentence, in the Results: “in both monkeys, a modest number of trials exhibited spontaneous vacillations.” Since decoded choices are ultimately a reflection of population firing rates, is there some feature of the population firing rate that was different on vacillated trials? Namely, were firing rates lower? If overall population firing rates were similar between vacillated trials and regular trials, is there a subset of neurons that changes firing rate? It would be good to understand if/how neural firing rates in both single cells and the population give rise to vacillations*.

This is a great question, and we have performed and added a new analysis to answer it. We both examined free choice vacillations and changes of mind induced by barrier changes, specifically, trials which were free throughout but in which a barrier changed and the decoded choice switched from one side to the other. In both cases, firing rates are sometimes higher and sometimes lower than average. However, these two kinds of changes-of-mind did not exhibit the same patterns of neural activity. We have now added the following to the Results (with further details in the Methods):

“Second, we asked whether there was a population signature during changes of mind. […] so this firing rate difference may reflect the time course of motor preparation rather than a difference between spontaneous vacillation and induced changes of mind.”

We sincerely thank the reviewers for motivating us to look at the firing rates this way, and we think it has added an important piece to the story.

*In the Results, the authors state: “to our knowledge, observations of such spontaneous vacillations when presented with a free choice—without changing external evidence—have not previously been described”. This claim appears unsubstantiated. Vacillations could simply be measurement error in the neural trajectory. Indeed, in 40% of the datasets, there was no significant difference in the number of vacillated trials between free and forced choice conditions*.

To clarify, all 5 of the original datasets showed more vacillations in free choice trials than in forced choice trials. Only 3 of the 5 datasets individually achieved statistical significance. The 2 datasets that were not significant still provide evidence in favor of our conclusion; they are just not individually significant. In addition, both of the two new datasets (added in this revision) also include more vacillations in free choice trials than in forced choice trials, and both these datasets were individually statistically significant.

This is a central point of this work, and we thank the reviewers for noting that it was confusing. We have now clarified this sentence, added a pooled statistic, and updated to include the additional data (in the subsection headed “Spontaneous changes of mind (vacillations) are present in the neural activity”):

“Vacillation trials were more common among free-choice trials than forced-choice trials in every one of the 7 datasets (p<10^-9^ pooled; χ^2^ 2x2 contingency test; Figure 4, left). This difference in frequency also reached statistical significance for 5/7 datasets when considered individually (p<0.05, χ^2^ 2x2 contingency test).”

*Discussion, first paragraph: Consider comparing these results with the work of Stanford and colleagues on saccades (Nat Neurosci 13:379, 2010; J Neurosci 33:16394, 2013), which also demonstrates competition between motor plans within the same task, from quick choices to slower trials in which the initially dominant motor plan is suppressed and overtaken by the alternative one (‘changes of mind'). Although that work is based on trial-averaged data, it tightly relates those motor planning patterns to behavior, and to RT in particular*.

We agree that this is relevant, and now mention this idea and these references in a new section of the (largely rewritten) Discussion. This portion reads:

“Other studies have presented two targets that were disambiguated only after the Go cue; in those circumstances, two competing motor plans have been observed both in the saccadic system (20; 44) and for reaching movements (16; 17). Here, however, competing motor plans seem unlikely to have interfered with the decoder on most trials: competing motor plans would tend to produce a decoded choice of intermediate magnitude, which was not typical in our data.”

*Discussion, second paragraph: “our data seem at odds with standard integrate-to-bound models.” In what way are they incompatible? Most of the neurons in*
Figure 2—figure supplement 1
*differentiate one choice from the other in less than 200 ms relative to the onset of the maze, showing an initial rise and differentiation that resemble the data from saccade-based tasks. (Responses are much more variable when aligned on movement onset, but that is understandable given the dependencies of PMd and M1 neurons on kinematic parameters). Second, such rise-to-threshold models have been used successfully in a variety of choice tasks, not just in random-dots tasks ’when information is delivered slowly.‘ But if ‘ramping’ is understood simply as an increase in motor planning activity, then the classic ramping associated with saccades is indeed very quick, on the order of 200 ms (Science 274:427, 1996; Neuron 76:616, 2012). Again, the work of Stanford and colleagues illustrates that a scheme in which alternative motor plans compete to reach a threshold works well even when decisions are urgent and choices quick. The work of Cisek and colleagues is consistent with such competition too, but over much longer time scales (Neuron 45:801, 2005; J Neurosci 29:11560, 2009; Neuron 81:1401, 2014). Thus, the rate at which sensory information is delivered or perceived is orthogonal to this issue of how to characterize motor-planning activity*.

This is a good argument. After careful consideration, we have removed our discussion of integration to bound. We agree with the reviewers that our data do not bear on this issue. We sincerely appreciate this most helpful feedback.

*The following statement in the Results: “the slow RTs on these trials were not due to re-planning, but instead involved a hesitation to execute.” The high RTs on untaken switch trials (*Figure 4*) is reminiscent of a phenomenon known as saccadic inhibition, in which a the appearance of an irrelevant distracter slows down a planned saccade to an unambiguous, unique, predictable target (J Cogn Neurosci 14:371, 2002; J Neurosci 31:12501, 2011; Vision Res 69:32, 2012). This is as if a sensory event (a change in a barrier, in this case) transiently interrupted and delayed an ongoing motor plan*.

This is a great comparison; thanks for pointing us to these references. The new text reads (in the Discussion):

“On hesitation trials (slowed RT when a new option was presented but not exercised) […] As we learn more about each process, a more mechanistic comparison may become possible.”

Figure 2*: what are the dots on the decoder trajectories in this case? *

We have now added “Small dots at last decoded time point” to the Figure 2 legend. We include these dots to make it easier to see when a trajectory ends in a spot that is crowded with other trajectories. These should not be confused with the larger dots on the traces in Figure 4; those indicate a change in barrier, with the color indicating which side the change made more favorable.

*Free-to-forced trials: it may be nice to show the decoder in these trials before and after the change in the maze*.

This is a great idea, and contributes to our ability to verify that our decoder can track fast changes. These plots now form Figure 3 and Figure 3—figure supplement 1. In addition, we added some quantification (in the third paragraph of the subsection “Early decision activity relates to behavior“):

“Second, we wished to confirm that our readout could reflect the monkeys’ choices at fast timescales. […] Thus, it was possible for the decoder to detect transient choices.”

Forced-to-free trials: how do we explain trials without a detectable change of mind?

In our rewritten Discussion, we now include a substantial section on this important topic:

“While neural responses to forced and free choices were generally similar […] that can continually reevaluate both what act to make and whether to make it now.”

What happens during unsuccessful trials?

On most unsuccessful trials, the monkey clipped a barrier on his way to the target. Occasionally, he lifted his hand a little too far from the screen (we set a limit on this distance), or made a saccade before the Go cue. Other failure modes, not holding the target long enough, making too slow of a reach, reaching too early, were rare. He essentially never did something nonsensical, such as trying to reach to a blocked target. By far the most common reason for failure was grazing the corner of a barrier. We now discuss this in the Methods:

“Errors for both novel and familiar mazes were almost always due to grazing the corner of a barrier along the correct path; other types of errors were much less common, and reaching toward a blocked target or into a dead end was extremely rare.”

We carefully considered whether there might be anything to learn from unsuccessful trials. One type of trial was of possible interest: “impossible trials” in which both targets were inaccessible. The monkey will necessarily be unsuccessful on such a trial, but we can ask what his neural activity does while he waits and perhaps hopes that a barrier change will make a target accessible. Most of the datasets show consistently weak planning for impossible trials, while one showed frequent strong vacillations. But since we cannot verify these decoded choices (no reach was made), we have decided to omit this unverifiable result.

*The title is compelling, but the three phenomena it lists*—*vacillation, indecision and hesitation—are not clearly distinguished in the manuscript. It would be helpful if the manuscript described them better.*

*For instance, the definition of a vacillation (or change of mind, but better stick to one word) is clearly explained in the Methods, but it would be useful if the main text mentioned that they were identified as big swings in the decoded choice*.

This is now done in several places. E.g.:

“In order to search for fully covert processes such as vacillation (unprompted changes of mind)…” (In the subsection “Early decision activity relates to behavior”).

“Indeed, in both monkeys a modest number of trials exhibited such spontaneous vacillations… we identified trials in which the decoder initially produced a value indicating one choice, then underwent a large swing to indicate the opposite choice.” (In the subsection “Spontaneous changes of mind (vacillations) are present in the neural activity”).

“This suggests that the slow RTs on these trials were not due to re-planning / vacillation, but instead involved a “hesitation” to execute. That is, when presented with a new option, the monkeys hesitated to execute their seemingly still-valid motor plan.” (In the subsection “Hesitation and indecision”).

“… the monkey did appear to undergo such moments of indecision: when a choice had to be made immediately upon maze presentation, the RT was often longer for two possible choices than for one choice” (in the same subsection).

*Results, tenth paragraph: consider adding some clarification when mentioning “…on free-choice trials…”. Rather, the lower performance may reflect somewhat higher variability or complexity during free choices or something similar*.

We have now added some clarification: “Rather, the lower performance of the decoder may reflect some additional complexity during free choices, such as last-moment changes of mind” (in the subsection headed “Free choices”). The reviewers’ hypothesis about higher variability is a good guess, but does not appear to be the case, at least not in any coordinated fashion. We have inspected the trajectories in a high-dimensional data viewer (DataHigh; http://users.ece.cmu.edu/∼byronyu/software/DataHigh/datahigh.html), and the neural trajectories for free choice trials have similar variability to those for forced choice trials. Formal tests for unequal variance (“heteroscedasticity”) also turned up little of interest. Sometimes free had greater variance in a dimension, sometimes forced did, and it varied by dimension within a dataset. Since the variance story is complicated and largely uninformative, we have chosen to omit it.